# Deforestation-induced climate change reduces carbon storage in remaining tropical forests

Yue Li [1✉], Paulo M. Brando [1], Douglas C. Morton[2], David M. Lawrence [3], Hui Yang[4] & James T. Randerson [1]

Biophysical effects from deforestation have the potential to amplify carbon losses but are often neglected in carbon accounting systems. Here we use both Earth system model simulations and satellite–derived estimates of aboveground biomass to assess losses of vegetation carbon caused by the influence of tropical deforestation on regional climate across different continents. In the Amazon, warming and drying arising from deforestation result in an additional 5.1 ± 3.7% loss of aboveground biomass. Biophysical effects also amplify carbon losses in the Congo (3.8 ± 2.5%) but do not lead to significant additional carbon losses in tropical Asia due to its high levels of annual mean precipitation. These findings indicate that tropical forests may be undervalued in carbon accounting systems that neglect climate feedbacks from surface biophysical changes and that the positive carbon–climate feedback from deforestation-driven climate change is higher than the feedback originating from fossil fuel emissions.

[1] Department of Earth System Science, University of California, Irvine, CA, USA. [2] Biospheric Sciences Laboratory, NASA Goddard Space Flight Center, Greenbelt, MD, USA. [3] National Center for Atmospheric Research, Boulder, CO, USA. [4] Department of Biogeochemical Integration, Max Planck Institute for Biogeochemistry, Jena, Germany. ✉email: yue.li@uci.edu

Tropical forests store more than 200 Pg C in aboveground live biomass[1–3]. Climate warming has the potential to contribute to a positive feedback that causes tropical forests to lose carbon, making it more difficult to stabilize the Earth's climate[4,5]. Over the past several centuries, the expansion of agriculture in tropical regions has contributed to widespread losses of tropical forest on multiple continents[6–8]. The rate of tropical forest loss accelerated in the 1970s (e.g., refs. [9,10]), and by the 1990s, direct carbon emissions from tropical deforestation were at record high levels, with estimates ranging between 0.8 and 2.2 Pg C yr$^{-1}$ (refs. [11,12]). Since the 2000s, rates of tropical deforestation have slowed, contributing to an overall decline in the global carbon flux from land-use change (e.g., from 1.9 Pg C yr$^{-1}$ in 1997 to 1.0 Pg C yr$^{-1}$ during 2010–2019, ref. [11,13]). Many tropical forests continue to lose carbon in hot-spot regions[14], however, as a consequence of increasing impacts from fire and other drivers of forest degradation[15,16]. Altogether, cumulative carbon emissions from tropical deforestation and other land-cover changes in the tropics over the past several centuries are comparable to the current aboveground vegetation carbon stock[17].

Apart from contributing to the build-up of atmospheric $CO_2$, tropical deforestation alters surface biophysical properties, contributing to decreases in evapotranspiration and surface roughness and increases in albedo when forests are replaced by grasslands and crops[18–21]. The influence of these biophysical changes has been long appreciated by regional and global climate modeling communities[22–27] and are known to contribute to regional warming and drying[28–32], changes in regional atmospheric circulation and moisture convergence[33,34], and longer-range teleconnections[35–37]. Precipitation responses to tropical deforestation also likely depend on the magnitude and spatial structure of the deforestation pattern. On broader spatial scales, decreases in evapotranspiration may weaken recycling along transport pathways delivering moisture to tropical forests from ocean source regions[38,39]. At a finer spatial scale, if deforestation contributes to a heterogeneous distribution of surface roughness and atmospheric heating, rainfall may increase in cleared areas and downwind of deforestation patches[40,41].

Contrasting climate responses to deforestation across different tropical continents have been found in model simulations[30,42,43] and are summarized in a recent review[32]. Studies using climate models agree generally that Amazonian deforestation has the strongest climate impacts, causing regional warming and decreases in precipitation. The regional climate response to deforestation in tropical Africa and Southeast Asia is weaker in magnitude, likely as a consequence of different forms of land-cover change, different climate baseline states, spatial patterns of deforestation, and geographical differences in topography and proximity of forests to nearby ocean regions[44]. For instance, the higher precipitation sensitivity to local surface drying in the Amazon, as compared to other tropical continents, has been recently attributed to different contributions of local evaporative recycling to precipitation in the baseline climate[45].

An important question when considering the net impact of deforestation on the Earth system is whether deforestation-induced changes in regional climate influence the local environment for remaining forests, making it either easier or more difficult for these forests to grow. Forest loss that causes regional warming and drying, for example, has the potential to contribute to positive carbon–climate feedback because higher air temperatures may reduce photosynthesis[46,47] and increase autotrophic respiration, leading to lower levels of net primary production and forest cover in nearby areas. Warming and drying also promote drought and wildfire[15,48], which greatly increases the risk of regional forest dieback and the associated loss of the aboveground

biomass[49]. An ensuing climate-tipping point, once triggered, may cause local ecosystems to move toward an alternate stable state[5,32,50,51], in which grass plant functional types are dominant, wildfires are prevalent, and carbon stocks are considerably reduced. Negative feedback, in contrast, may occur if loss of forest cover contributes to changes in atmospheric circulation that increase regional rainfall[40] or diffuse light[52]. While previous studies have explored deforestation edge effects[53,54] on carbon storage in nearby patches from changes in canopy microclimate and fire risk, much less work has examined the carbon consequences of regional-to-continental-scale changes in climate. To determine the magnitude and sign of these larger-scale interactions, here we quantify the influence of deforestation-driven climate change on the carbon storage of forests across different tropical continents. This is important because natural climate solutions are gaining attention as a possible mechanism to slow climate warming. In forest carbon offset programs, a critical need is to provide an accurate estimate of the carbon and climate benefits of a land management action (e.g., avoided deforestation), thus enabling a more effective valuation of the carbon credits issued for a specific project.

In this work, we estimate the biophysical impacts of deforestation on aboveground vegetation carbon stocks by combining deforestation-induced changes in annual mean rainfall and air temperature derived from an idealized global deforestation experiment (deforest–globe) that is part of the Land Use Model Intercomparison Project (LUMIP[55]) of phase 6 of Coupled Model Intercomparison Project (CMIP6), with empirical relationships between climate and aboveground biomass storage derived from contemporary satellite observations. Our analysis compares the relative magnitude of the biophysical carbon cost to the direct aboveground biomass loss from tropical deforestation across three different continental regions (Amazon, Congo, and the maritime continent in tropical Asia). We define the biophysical carbon cost as the additional loss of carbon driven by deforestation-induced climate change. We also report the carbon–climate feedback parameter, gamma (defined as the cumulative carbon loss at each location for a 1 °C increase in surface air temperature)[56] driven solely by the biophysical climate effect of tropical deforestation and compare it to more traditional estimates of gamma derived from radiative effects of increasing $CO_2$.

## Results

**Deforestation impacts on tropical climate across continents.** The multimodel mean estimates (obtained from eight fully coupled Earth system models, ESMs[57–64], Table 1) of the tropical climate response (Fig. 1) to idealized deforestation (Supplementary Fig. 1) show that many areas across the tropics experience warming and reduced rainfall in response to deforestation. Decreases in rainfall occur across almost all of the Amazon, in the western half of the Congo basin and across the southern part of Borneo and the interior western part of New Guinea (Fig. 1a–c and Supplementary Fig. 2). These reductions are consistent with widespread declines in evapotranspiration in deforested areas, as shown by Boysen et al.[65], and are partly offset by precipitation increases in northern and eastern Africa. The largest decline of mean annual precipitation occurs in Amazonia, where a 59.0 ± 16.9% loss in biomass drives a significant precipitation decrease of 150 ± 105 mm yr$^{-1}$ (6.7 ± 4.7%), averaged across all eight models (Table 2). Precipitation reductions for the Congo and for the maritime continent (that is, islands in tropical Southeast Asia) are smaller in relative magnitude, with biomass losses of 50.8 ± 13.9% and 42.0 ± 10.2% resulting in mean annual precipitation decreases of 41 ± 56 mm yr$^{-1}$ (2.7 ± 3.7%) and

**Table 1 A list of information of 8 Earth system models participating in the LUMIP idealized deforestation experiment.**

| Model name | Model center | Original resolution | Dynamic vegetation | Reference |
|---|---|---|---|---|
| BCC–CSM2–MR | Beijing Climate Center, China | 320 × 160 (1.125° × 1.125°) | No | Wu et al.[57] |
| CanESM5 | Canadian Centre for Climate Modelling and Analysis, Canada | 128 × 64 (2.8° × 2.8°) | No | Swart et al.[58] |
| CESM2 | National Center for Atmospheric Research, USA | 288 × 192 (1.25° × 0.94°) | No | Danabasoglu et al.[59] |
| CNRM–ESM2-1 | Centre National de Recherches Météorologiques, France | 256 × 128 (1.4° × 1.4°) | No | Séférian et al.[60] |
| IPSL–CM6A–LR | Institut Pierre-Simon Laplace, France | 144 × 143 (2.5° × 1.27°) | No | Boucher et al.[61] |
| GISS-E2-1-G | NASA Goddard Institute for Space Studies, USA | 144 × 90 (2.5° × 2°) | No | Kelley et al.[62] |
| UKESM1-0-LL | Met Office Hadley Centre, UK | 192 × 144 (1.87° × 1.25°) | Yes | Sellar et al.[63] |
| MPI-ESM1-2-LR | Max Planck Institute for Meteorology, Germany | 192 × 96 (1.9° × 1.9°) | No | Mauritsen et al.[64] |

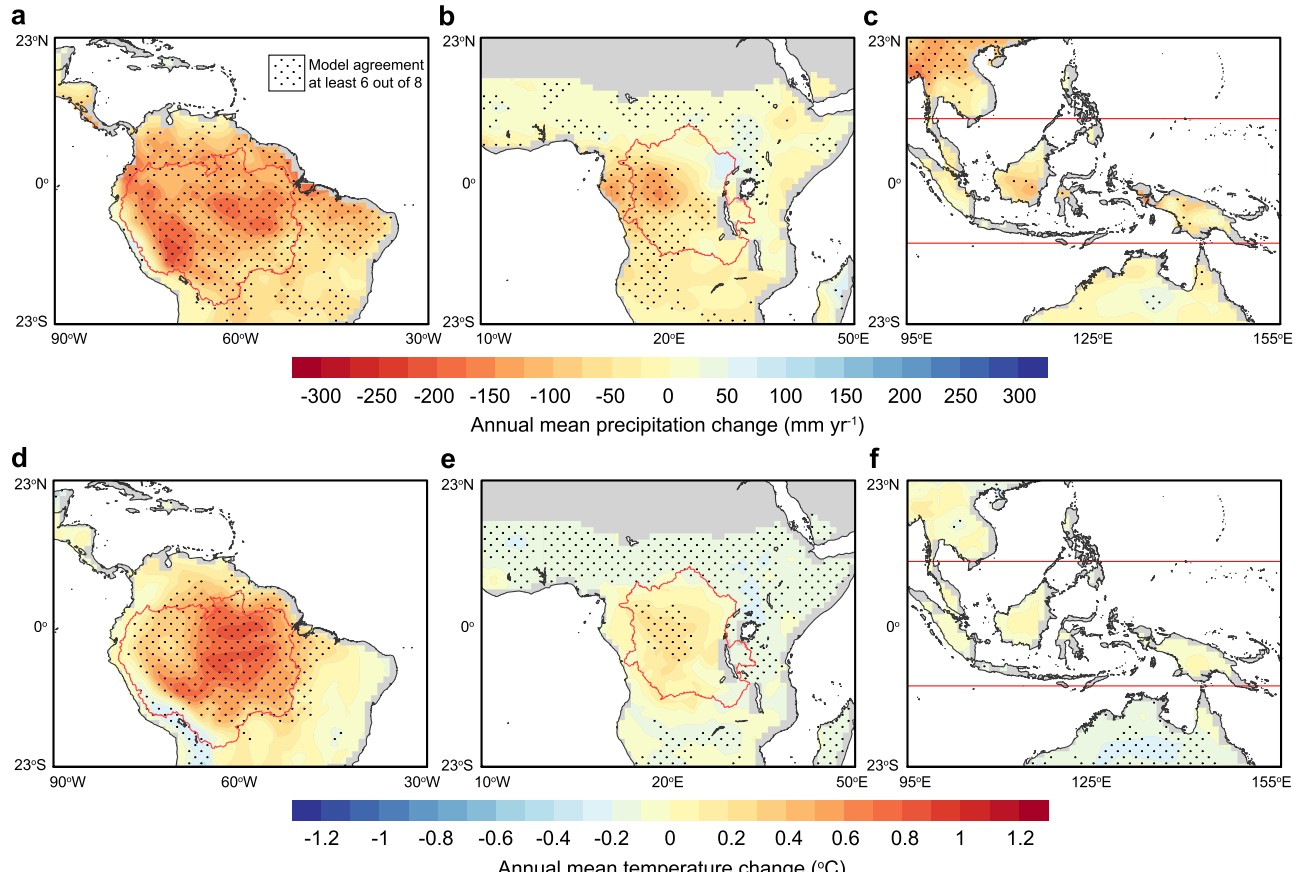

**Fig. 1 Biophysical impacts of idealized deforestation on rainfall and temperature in three tropical regions.** Changes in mean annual rainfall and air temperature in **a**, **d**, South America, (**b**, **e**, Africa and **c**, **f**, Southeast Asia. The changes were computed as the difference between the average of last 30 years from the LUMIP deforest–globe and piControl experiments (see "Methods"). The dotted area indicates the model agreement, with at least six out of eight models agreeing on the sign of the climate responses. Information on the eight models is listed in Table 1.

$38 \pm 58$ mm yr$^{-1}$ (1.3 ± 2.0%), for the Congo and the maritime continent, respectively. Model agreement with respect to the direction of the precipitation response to deforestation is high in South America, with at least six of the eight models showing decreases in precipitation across most of the Amazon. Model agreement is lower in the eastern part of the Congo and across tropical Asia, where the magnitude of the multimodel mean change is also smaller relative to background precipitation levels.

In response to idealized deforestation, mean annual air temperature increases significantly in the Amazon by $0.5 \pm 0.5$ °C for the multimodel mean, with smaller and more variable cross-model responses in the Congo ($0.1 \pm 0.5$ °C) and tropical Asia ($-0.1 \pm 0.2$ °C) (Fig. 1d–f and Table 2). The continental differences in the mean annual temperature response

have long been recognized from early climate model simulations of tropical deforestation[30] and are explained by tradeoffs between declining surface net radiation (which causes cooling) and reductions in evapotranspiration and surface roughness (which causes surface warming). In the Amazon, latent heat decreases by $6.9 \pm 3.5$ W m$^{-2}$ from deforestation whereas net surface radiation declines by $4.6 \pm 2.5$ W m$^{-2}$ (Supplementary Table 1). In contrast, the latent heat declines by $3.4 \pm 4.5$ W m$^{-2}$ and $2.1 \pm 3.8$ W m$^{-2}$ for the Congo and the islands of tropical Asia, respectively, relative to declines in net radiation of $3.3 \pm 2.9$ W m$^{-2}$ and $1.4 \pm 1.9$ W m$^{-2}$. The weaker warming response in the Congo may be driven by a smaller cloud response (that is, a smaller decrease in cloud cover, Supplementary Fig. 3) in Africa where the diurnal temperature range changes by a

**Table 2 Impacts of idealized deforestation on climate and aboveground biomass (AGB) in the tropics[a].**

| Region | Tree cover change (%) | Forest biomass change (%) | Relative precipitation change (%) | Precipitation change[c] (mm yr⁻¹) | Temperature change (°C) | Total forest AGB loss[d] (Mg C ha⁻¹) | Estimated AGB loss from the biophysical feedback (Mg C ha⁻¹) | Estimated AGB loss from the biophysical feedback (%) |
|---|---|---|---|---|---|---|---|---|
| Amazon[b] | −44.7 (6.0) | −59.0 (16.9) | −6.7 (4.7) | −150 (105) | 0.5 (0.5) | −98.3 (13.2) | −5.0 (3.6) | 5.1 (3.7) |
| Congo | −38.7 (8.8) | −50.8 (13.9) | −2.7 (3.7) | −41 (56) | 0.1 (0.5) | −75.5 (16.7) | −2.9 (1.9) | 3.8 (2.5) |
| Tropical Asia | −31.2 (8.9) | −42.0 (10.2) | −1.3 (2.0) | −38 (58) | −0.1 (0.2) | −62.4 (17.8) | −0.3 (2.0) | 0.5 (3.2) |

[a]In parentheses, we present 1 standard deviation (SD) of the mean across models, with the uncertainty of biophysical AGB loss propagated from that of the precipitation and temperature using the equation for observations in Table 3.
[b]Amazon and Congo are defined as land grids within the basin map, and tropical Asia is defined as the land grid cells within 10°S–10°N, 95°E–155°E.
[c]Precipitation change was computed as the product between multimodel average relative rainfall change (%) and climatological rainfall observations at each continent, to avoid the influence of simulated rainfall bias on the deforestation impacts on rainfall for each model.
[d]Total forest carbon loss was estimated as the product between the tree cover change in LUMIP deforest–glob simulations after 50 years and the observational tree cover–aboveground biomass relationship as shown in Supplementary Fig. 7.

smaller amount as shown by previous simulations[36,43]. An asymmetric cloud response between the Amazon and the Congo is seen in ESMs such as CanESM5, CESM2, MPI–ESM–1.2.0, and UKESM1-0-LL, as shown in Supplementary Fig. S6 in ref. [65]. The smaller latent heat decline for the tropical Asian islands is expected since climatological precipitation is much larger than in the other two regions, which means that evaporation is less frequently water-limited. Further, the temperature over the maritime continent is more tightly controlled by the surrounding ocean.

**The sensitivity of tropical aboveground biomass to climate.** To assess the response of tropical aboveground vegetation carbon storage to the biophysical climate effects of deforestation, we developed an empirical relationship between tropical aboveground biomass (AGB) from the ESA–CCI BIOMASS project[3] and climate observations (see "Methods"), drawing upon the spatial variability of biomass and climate across different tropical regions (Fig. 2a). We used all resampled 1-degree grid cells within 23°S–23°N (including forests and grasses) in our analysis except for desert regions with mean annual precipitation less than 100 mm yr⁻¹ and grid cells with a land fraction less than 50%. Across the tropics, drier areas also have more spatial variation in mean annual temperature, with lower AGB observed in regions with higher mean annual temperature (Fig. 2a). Both the Congo and the Amazon are closer to a 1500 mm yr⁻¹ threshold that separates tropical forests and savannas[66], suggesting forests in these regions are closer to a climate-tipping point. The climate–AGB relationship derived from the spatial variation in observations reflects the long-term evolutionary and adaptive responses of the terrestrial ecosystem to climate, and implicitly includes processes such as fires, deforestation and drought effects on tree mortality[66]. The CMIP6 multimodel mean captures the general spatial sensitivity of AGB to precipitation and temperature variations (Fig. 2b) despite the simulated AGB by models are biased a bit low in areas with a high fraction of forests (Supplementary Fig. 4).

We first used a multiple linear regression model to estimate the AGB sensitivity to the spatial variations in temperature and precipitation, with the aim of combining this information with the climate impacts of deforestation to estimate the biophysical feedback effect of deforestation on AGB. Table 3 shows that observed mean annual temperature and mean annual precipitation explain 49% of the spatial variance in tropical AGB using the climate and biomass observations, with the AGB increasing by 3.4 Mg C ha⁻¹ for 100 mm yr⁻¹ increase in precipitation (8.2% per 100 mm yr⁻¹). Similarly, AGB declines by 0.32 Mg C ha⁻¹ for every 1 °C increase in mean annual temperature (−0.8% per °C) (Table 3). This larger AGB sensitivity to precipitation is in contrast to a recent report[47] that finds that the tropical forest AGB is more sensitive to maximum temperature (−5.9% per °C) rather than rainfall (2.4% per 100 mm yr⁻¹). When comparing our findings to those in the previous study, it is important to note our regression is also derived from non-forest biomes in the tropics, including savannas and grasslands (Supplementary Fig. 5). Therefore, the climate gradient that we consider is larger, and rainfall therefore has a more significant role in defining the transition from forests to savannas.

To tailor a climate-biomass statistical model for use in wetter areas where tropical forests are dominant, we used a moving window on climatological annual rainfall (see "Methods"). This analysis indicates that tropical AGB rainfall sensitivity is the largest between 1500 mm yr⁻¹ and 2000 mm yr⁻¹ (Fig. 2c), collocated with the average rainfall climatology in the Congo basin and the southern and eastern parts of the Amazon. This

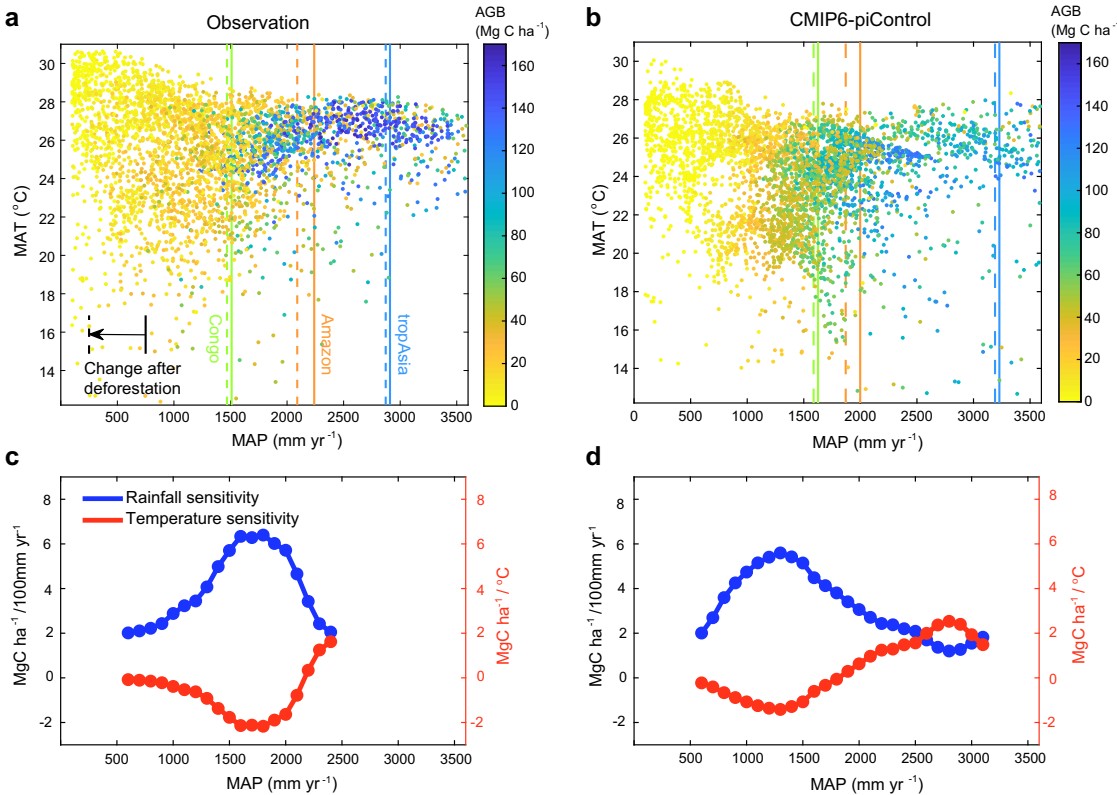

**Fig. 2 Aboveground biomass (AGB) relationships with mean annual temperature (MAT) and precipitation (MAP) derived from both observations and CMIP6 models. a**, **b** AGB from satellite data (ESA–CCI) and from CMIP6–piControl experiment averaged in climate space of MAT and MAP. AGB from CMIP6 was computed from the total vegetation carbon multiplied by a mean ratio between the aboveground and total biomass weighted by the tree cover fraction (see "Methods"). **c**, **d** Statistical sensitivity of AGB to MAP and MAT for observation-based data and CMIP6 simulations, respectively. The sensitivity of AGB to MAP and MAT was estimated from a regression within each moving window spanning ±500 mm yr$^{-1}$ and centered at each MAP level from 600 to 3100 mm yr$^{-1}$. The dotted curves indicate the range over which the regressions were significant at $P <0.001$. No significant relationships were found for the observations above a MAP of 2400 mm yr$^{-1}$. Vertical lines in panels a and b represent mean precipitation levels for each region before (solid) and after (dashed) the deforestation.

**Table 3 Overview of the statistical model of observed and simulated aboveground biomass (AGB) in relation to mean annual precipitation (MAP) and temperature (MAT).**

|  | a*100 | b | R$^2$ | RMSE | δ$_{MAP}$ | δ$_{MAT}$ |
|---|---|---|---|---|---|---|
| Observations[a] | 3.4 | −0.32 | 0.49 | 32 Mg C ha$^{-1}$ | 8.2% /100 mm yr$^{-1}$ | −0.8%/°C |
| CMIP6 mean | 3.2 | −0.04 | 0.60 | 23 Mg C ha$^{-1}$ | 6.9% /100 mm yr$^{-1}$ | −0.09%/°C |

[a]Equation: AGB = a*MAP + b*MAT + ε. The units are mm yr$^{-1}$ for MAP, °C for MAT, and Mg C ha$^{-1}$ for AGB. δ$_{MAP}$ and δ$_{MAT}$ indicate the relative AGB sensitivity to MAP and MAT (in percentage), computed as the relative value of the coefficients a and b to the observed/simulated AGB averaged for the whole tropical region. RMSE denotes the root mean square error.

suggests that the aboveground vegetation carbon storage is more sensitive to rainfall changes in tropical Africa and South America than in tropical Asia. An AGB spatial sensitivity analysis with two other satellite AGB products[1,2] shows consistent results to those derived with the ESA–CCI product (Supplementary Fig. 6).

Within the CMIP6 models, the multiple linear regression explains 60% of the spatial variability in AGB (Table 3). The sensitivity of AGB to precipitation within the CMIP6 models is similar to the observations (6.9% per 100 mm yr$^{-1}$) but the temperature sensitivity is considerably lower (−0.1 % per °C) (Fig. 2b, d).

We also explored the impact of other or more specific environmental factors such as vapor pressure deficit (VPD), mean annual maximum temperature, the seasonality of temperature and precipitation, and precipitation during the driest three months (see "Methods"). These metrics are thought to be physiologically meaningful for tropical forest growth[47,67].

We found that these physiological environment factors have a high spatial collinearity with the mean annual temperature and precipitation ("Methods") and, therefore, their capability to explain the spatial variation of observed AGB and simulated aboveground vegetation carbon is comparable to that of mean annual temperature and mean annual precipitation (Supplementary Tables 2 and 3). Consequently, for simplicity, we used the AGB sensitivity solely to mean annual temperature and mean annual precipitation (Fig. 2c) to quantify the impacts of deforestation-induced changes in regional climate on tropical AGB.

**AGB costs of deforestation-driven changes in climate.** To estimate the carbon costs of deforestation-driven changes in surface biophysics, we combined the climate changes from the LUMIP idealized deforestation experiment (Fig. 1) with the

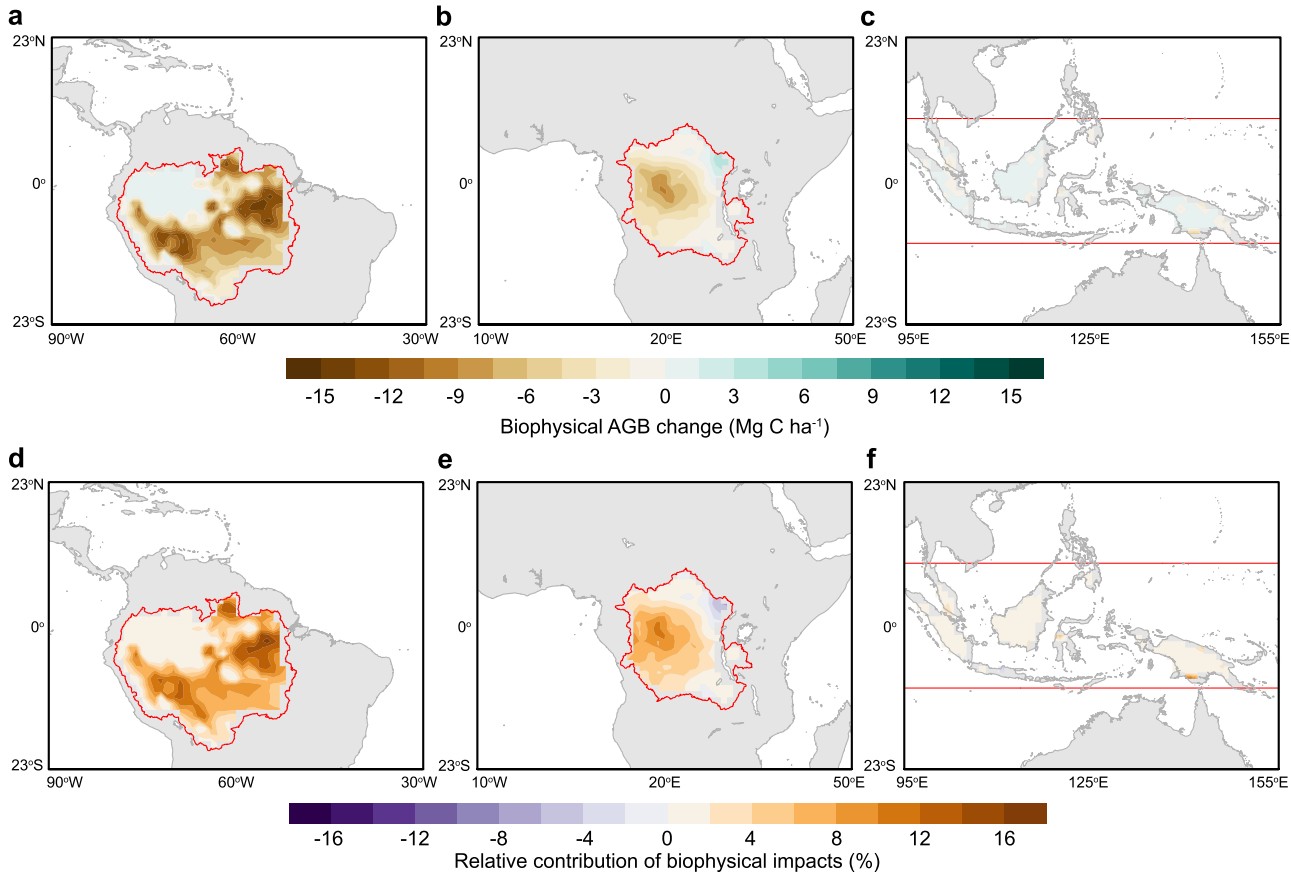

**Fig. 3 Biophysical impacts of deforestation on aboveground biomass (AGB) in the tropics. a–c** Shows biophysical AGB changes of deforestation estimated from the product of deforestation-induced changes in climate (mean annual precipitation and mean annual temperature), and the observational sensitivity of the AGB to precipitation and temperature shown in Fig. 2c. **d–f** Shows the relative change of (**a–c**) as a percent of the estimated direct AGB loss from deforestation (see "Methods").

moving window climate–AGB relationship we derived from the observations (Fig. 2c). Warming and drying from deforestation contributed to an AGB loss of $5.0 \pm 3.6$ Mg C ha$^{-1}$ in the Amazon, $2.9 \pm 1.9$ Mg C ha$^{-1}$ in the Congo, and $0.3 \pm 2.0$ Mg C ha$^{-1}$ for the islands in tropical Asia (Table 2). The satellite observations indicate that AGB decreases by about 19–22 Mg C ha$^{-1}$ per 10% decrease in tree cover fraction in the tropics (Supplementary Fig. 7). Using this relationship derived from the observations and the tree cover fraction changes from the LUMIP models, we estimated that the direct AGB carbon losses due to deforestation in the idealized deforestation experiments are $98.3 \pm 13.2$ Mg C ha$^{-1}$, $75.5 \pm 16.7$ Mg C ha$^{-1}$, and $62.4 \pm 17.8$ Mg C ha$^{-1}$ in the Amazon, Congo, and tropical Asia, respectively (Table 2). These estimated AGB carbon losses are larger than that simulated explicitly in the ESM simulations (Supplementary Fig. 1) as the model simulations of the aboveground vegetation carbon stocks are smaller than the satellite observations due in part to a lower atmospheric $CO_2$ level in the preindustrial era (Supplementary Fig. 4). Expressed relative to the direct AGB loss in the deforestation experiment, the biophysical effects of deforestation contribute to $5.1 \pm 3.7\%$ (that is, $(-5.0 \pm 3.6$ Mg C ha$^{-1})$/$(-98.3$ Mg C ha$^{-1}$)) of additional AGB loss in the Amazon, $3.8 \pm 2.5\%$ of additional loss in the Congo, and $0.5 \pm 3.2\%$ in tropical Asia.

Figure 3 shows the spatial pattern of the biophysical effects of deforestation on AGB and their percent contribution to total biomass loss related to the direct effects of deforestation. Despite the widespread decline in rainfall and warming, the largest AGB

loss due to biophysical feedback occurs in the eastern Amazon where the additional AGB loss is as high as 14 Mg C ha$^{-1}$ (17%) (Fig. 3a, d). There is no additional AGB loss predicted for the northwestern Amazon due to its high baseline precipitation level that reduces the sensitivity of AGB to changes in precipitation or temperature (Fig. 2c). Biophysical effects also amplify the AGB loss in the central Congo Basin by up to 9 Mg C ha$^{-1}$ (11%) but do not lead to any additional AGB loss in tropical Asia as a consequence of its high precipitation baseline (Fig. 3b, c, e, f). Further separation of the effects of deforestation-driven changes in temperature and precipitation indicates that it is the precipitation response that controls the spatial pattern of the biophysical carbon costs (Fig. 4), highlighting the importance of water stress in regulating tropical deforestation-driven AGB changes.

By applying the relationship that we obtained from LUMIP simulations and the satellite AGB observations, we can estimate the influence of past deforestation during the historical era on regional climate and the associated additional carbon losses. For South America, the mean primary forest fraction declined by 11.5% in the Amazon basin from 1850 to 2015, according to the Land Use Harmonization (LUHv2h) dataset[68]. Using the information from the LUMIP simulations, this decline would translate into a regional precipitation decrease of about 38.6 mm yr$^{-1}$ and a temperature increase of about 0.13 °C. Multiplied by the observed AGB–climate sensitivity (Fig. 2c), these biophysical climate effects (mainly from rainfall reduction) associated with deforestation would be equivalent to an additional

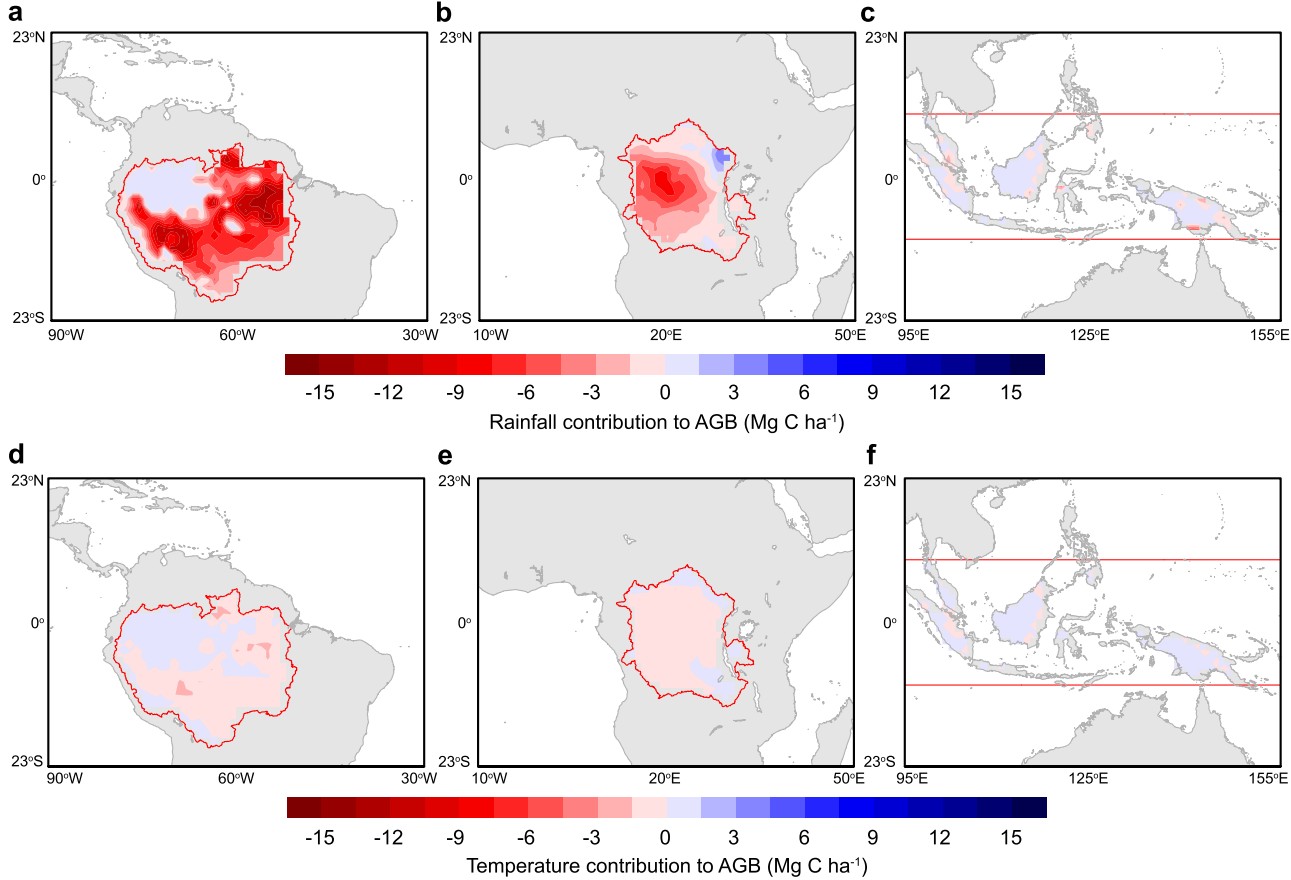

**Fig. 4 Contribution of rainfall and air temperature change to the total biophysical aboveground biomass (AGB) loss associated with deforestation.**
**a–c** Shows precipitation contributions to the biophysical AGB costs of deforestation across the three different tropical regions. **d–f** Shows temperature contributions to the biophysical AGB costs of deforestation. The partitioning is based on results shown in Fig. 3a–c but separates the contributions from the deforestation-induced change in annual mean precipitation and temperature. The relative contribution of precipitation change to biophysical AGB loss varies from 80 to 100% in the Amazon and from 85 to 100% in the Congo.

AGB loss of 741 Tg C (Supplementary Table 4). In Africa, 8.4% loss in primary forest fraction was estimated to cause regional reduction in rainfall by 8.8 mm yr$^{-1}$ and warming by 0.02 °C, resulting in a cumulative biophysical AGB loss of about 200 Tg C from 1850 to 2015 (Supplementary Table 4).

**Deforestation-driven climate–carbon cycle feedback parameter.**
Climate warming is projected in CMIP5 and CMIP6 models to reduce tropical land carbon storage, which is described by a γ parameter that is often negative in sign for tropical terrestrial ecosystems (see Fig. 6.22 in refs. [69] and [70]). Here we estimated γ from the LUMIP models by combining deforestation-driven temperature and carbon stock changes and compare this estimate of γ (i.e., $\gamma_{AGB}^{def,biophys}$, see "Methods") with estimates derived from $CO_2$-driven climate change (i.e., $\gamma_{AGB}^{CO2}$, see "Methods") for the same set of CMIP6 models. Similar to previous conclusions regarding the response of land carbon storage to climate change[69], we find that the most negative $\gamma_{AGB}^{CO2}$ is in the Amazon, with smaller magnitude of $\gamma_{AGB}^{CO2}$ values in the Congo and across the islands in tropical Asia (Fig. 5a–c). It should be noted that this $\gamma_{AGB}^{CO2}$ calculation does not account for any biophysical climate effects of land use and land-cover change[71]. We apply a similar approach to calculate a new deforestation-driven climate–carbon feedback parameter $\gamma_{AGB}^{def,biophys}$ by normalizing the biophysical carbon costs of tropical deforestation by the deforestation-caused warming (see "Methods"). Figure 5d–f show that the $\gamma_{AGB}^{def,biophys}$ is, in general,

more than twofold higher than the $\gamma_{AGB}^{CO2}$ derived from the idealized $CO_2$ increasing experiments in the Amazon and Congo. This is because the biophysical effects of deforestation influence the aboveground vegetation carbon mainly through a regionally concentrated rainfall response, with the temperature response being much smaller. Our analysis on the biophysical carbon costs of tropical deforestation, therefore, suggests that the positive carbon–climate feedback from deforestation is fundamentally larger than the feedback originating from fossil emissions.

**Discussion**
Our analysis indicates that the biophysical climate effects of tropical deforestation, particularly as a consequence of precipitation reductions, add to committed carbon emissions by an extra 5.1 ± 3.7% in the Amazon and by an extra 3.8 ± 2.5% in Congo. The additional carbon losses occur as a consequence of remaining intact forests experiencing hotter and drier conditions that reduce carbon storage in aboveground biomass. These findings suggest that the value of avoided deforestation and forest degradation may be underestimated if current carbon assessment methodologies focus only on the direct carbon stock and emission changes associated with the land-use change[72,73]. The biophysical additionality identified here should be complementary to other co-benefits, including the potential of remaining forests to serve as a future terrestrial carbon sink in response to rising levels of atmospheric carbon dioxide and other global change drivers[74,75]. The regional differences in the deforestation biophysical changes

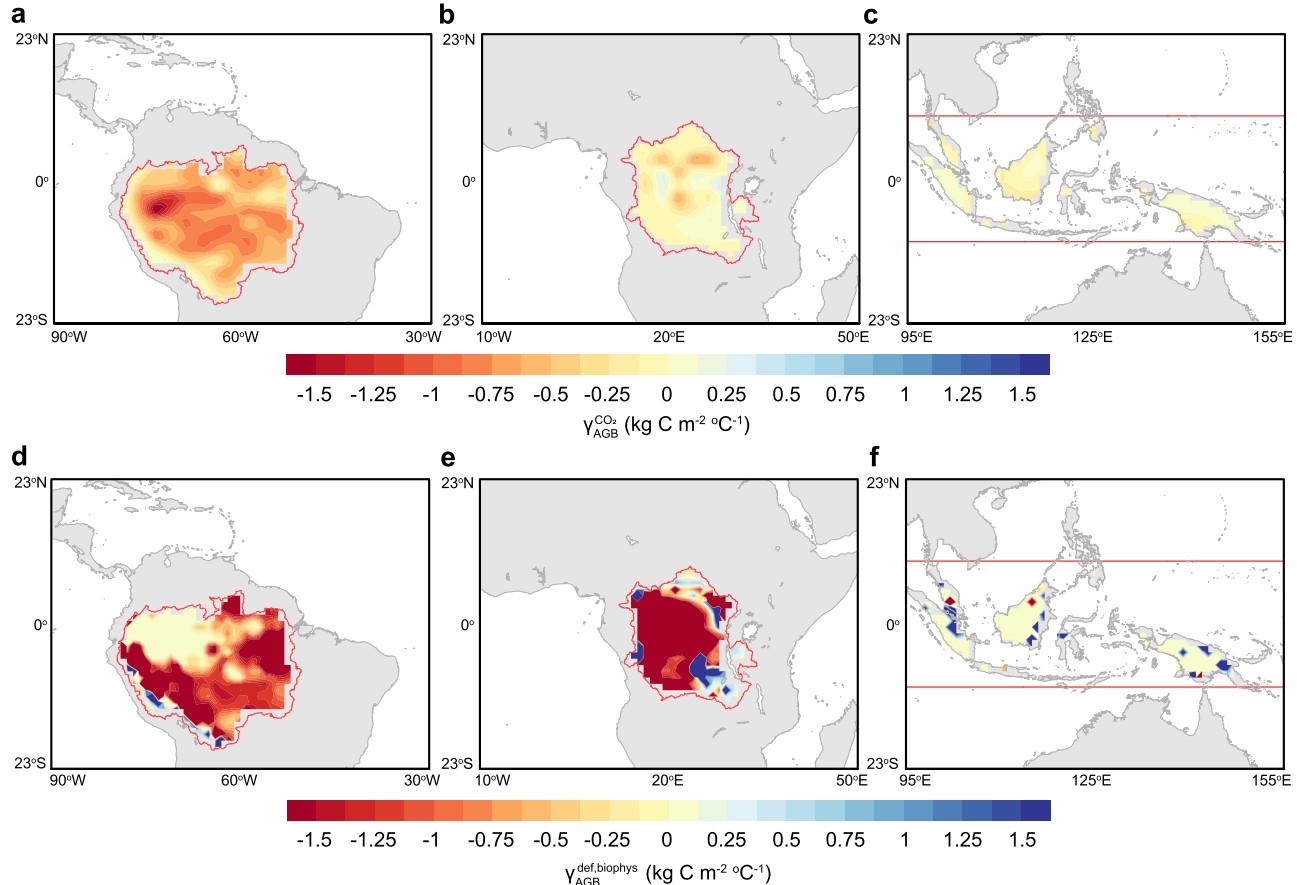

**Fig. 5 Spatial pattern of the $CO_2$-driven and deforestation-driven carbon–climate feedback parameters. a–c** Shows the climate–AGB feedback parameter ($\gamma$) from an idealized increasing of ~4 × $CO_2$ experiments ($\gamma_{AGB}^{CO2}$), and **d–f** shows the same quantity derived from the climate change derived from the LUMIP deforestation experiment and associated AGB losses ($\gamma_{AGB}^{def,biophys}$). $\gamma_{AGB}^{CO2}$ was computed as the difference of changes in AGB between the full and biogeochemical transient $CO_2$ increasing experiments normalized by warming under the full coupled warming as the following ref.[70]. $\gamma_{AGB}^{def,biophys}$ was computed as the ratio of the deforestation-driven climate impacts on the AGB relative to the deforestation-induced climate warming (see "Methods").

**Table 4 Additional benefits for avoiding deforestation or reforestation associated with the indirect local and regional effects of deforestation on aboveground biomass (AGB).**

| Mechanism | Impact of edge or climate feedback on AGB | | Citation |
|---|---|---|---|
| | **Mean** | **Range** | |
| Local edge effects (from changes in microclimate and fire) | 36% (Amazon)[a] | 25–56% | Junior et al.[54] |
| | 19% (Congo)[b] | 18–20% | Zhao et al.[77] |
| | 10% (Tropical Asia)[c] | 7–13% | Ordway and Asner[78] |
| Regional climate feedback (from changes in rainfall and temperature) | 5.1% (Amazon) | 1.4–8.8% | This study |
| | 3.8% (Congo) | 1.3–6.3% | |
| | 0.5% (Tropical Asia) | 0.0–3.7% | |
| Local edge + regional climate feedback | 41% (Amazon) | 26–65% | |
| | 23% (Congo) | 19–26% | |
| | 11% (Tropical Asia) | 7–17% | |

[a]Ratio between total gross carbon loss from edge effect and that from deforestation during 2001–2015.
[b]Ratio between carbon loss from edge effect and that from deforestation for scenarios 1–3 under the Representative Concentration Pathway 8.5 (RCP8.5) for the entire continent of Africa.
[c]Ratio of biomass loss has been reported to range from 16 to 30% according to forest sites observations in Sabah, Malaysian Borneo[78]. This ratio has been multiplied by an average fraction of forest fragmentation of 31% (Fischer et al.[79]), and further divided by the remaining forest fraction of 69% that could experience potential large-scale deforestation.

in forest carbon stocks that we uncover may also provide insight about a more equitable approach for assigning carbon credits in the context of Reducing Emissions from Deforestation and forest Degradation projects (REDD+) (e.g., ref.[76]) and other climate policy frameworks. An important next step in this context is to combine the deforestation biophysical climate effect identified here with AGB losses associated with local edge effects[54,77–79] in order to estimate an integrated indirect carbon benefit associated with avoided deforestation (or reforestation) projects in the tropics. A preliminary comparison of these two mechanisms and their total effects are provided in Table 4. In the Amazon, these indirect benefits of avoided deforestation may sum to be 41% (with a range of 26–65%) higher than carbon contained within the project boundaries.

The positive biophysical climate feedback of tropical deforestation (e.g., warming and reduced precipitation as revealed by previous single model-based studies[22–27] and our multimodel analysis) has important implications for assessing future climate risks of tropical moist forests[50,80]. From our analysis of LUMIP–CMIP6 simulations, idealized deforestation by 45% contributes to an Amazonian rainfall decline by $150 \pm 105$ mm yr$^{-1}$, which is comparable to the expected change in rainfall observed in idealized 1% $CO_2$ physiological effect experiments (that is, by $-175$ mm yr$^{-1}$ in response to a $4 \times CO_2$ increase for CMIP5, ref. [45]). If we presume that these two effects can be linearly combined that would mean a 325 mm yr$^{-1}$ reduction of precipitation for a climatology of about 2240 mm yr$^{-1}$ (Fig. 2). An annual mean precipitation threshold of 1500 mm yr$^{-1}$ has been identified as the typical hydrological boundary between tropical forests and savannas[66,81]. The total precipitation expected changes from $CO_2$ physiology and deforestation together suggest that the mean Amazonian precipitation may not cross the hydrological threshold necessary for a permanent transition to a savanna-like state in response to these drivers. However, in southern and eastern areas of the Amazon, where current precipitation levels are much closer to this tipping point, these combined effects may push ecosystems over this threshold and may be further amplified by the direct (radiative) effects of global climate change.

In contrast, deforestation-driven decreases in rainfall in the western Congo and in some areas of tropical Asia may be partially offset by the radiative and physiological effects of rising $CO_2$, which may increase precipitation in these regions as a consequence of interactions between surface biophysical changes and regional atmospheric circulation[45]. Nevertheless, the change of mean climate by deforestation (that is, less rainfall and warming), as revealed in our analysis of the LUMIP–CMIP6 simulations, still implies the increasing possibility of a lengthening dry season[82], an increasing amplitude of extreme drought events[83], and a higher likelihood of wildfires[5,10,15] in the tropics if deforestation continues in the future. More broadly, the high levels of uncertainty regarding how radiative, physiological, and land-cover change mechanisms influence tropical precipitation make it challenging to accurately predict climate-tipping points in the tropics.

The carbon–climate feedback parameter γ was initially proposed to measure the carbon cycle response to climate warming[56,70], irrespective of whether the warming originates from fossil fuel emissions or carbon emissions from land-use change. In past applications, γ has been found to be negative (i.e., a loss of carbon to the atmosphere for a 1 °C temperature increase) across tropical land ecosystems as a consequence of increases in ecosystem respiration and decreases in photosynthesis caused by warming[84]. As far as we are aware, past work has not estimated γ driven by warming from the biophysical effects of tropical deforestation. This warming is fundamentally different because it is associated with large changes in precipitation and other land surface variables in the tropics, including humidity and wind speed. Here, we find that the deforestation-driven $\gamma_{AGB}^{def.biophys}$ (i.e., the term considering the biophysical warming effect of deforestation on the carbon cycle in the tropics) could be twofold larger in the Amazon and Congo than that computed from the idealized $CO_2$ increasing experiment without consideration of land-use and land-cover change ($\gamma_{AGB}^{CO2}$, as derived from the Coupled Climate–Carbon Cycle Model Intercomparison Project experiments (C4MIP) for CMIP6, ref. [71]). This implies that the warming caused by biophysical effects of tropical deforestation has stronger impacts on nearby tropical terrestrial ecosystems than warming originating from global radiative forcing of the Earth system, once adjusted for the same change in temperature. Further analysis on the attribution of future tropical climate change to deforestation and $CO_2$ is needed for a better understanding of the role of tropical land use and land cover in the climate system. In future work, higher resolution model simulations may help to identify optimal locations for forest restoration efforts in order to offset precipitation declines from historic deforestation and maximize climate change mitigation from tropical reforestation efforts.

There are several key uncertainties associated with tropical deforestation and its biophysical impact on precipitation and temperature including whether or not precipitation and temperature are linearly or nonlinearly dependent on the amount of deforestation, masking of the precipitation response by internal climate variability, and the potential for the deforestation-driven climate changes to impact fires and therefore to feedback onto aboveground vegetation carbon stocks by means of changes in the disturbance regime. Further work is needed to explore nonlinearities in the carbon costs of tropical forest loss and the drivers of continental-scale differences. Experiments and analysis using high-resolution models could help refine the cloud and convection responses to deforestation. More research on local and remote teleconnections is also needed. Specifically, more work is needed by the Earth system community to understand how deforestation or reforestation in different regions (on individual continents) influence local and remote patterns of precipitation[38] and other aspects of near-surface climate. This may include further effort to develop a process-based evaluation framework for quantifying the impact of deforestation on regional climate that reconciles predictions from ESMs with long-term trends from satellites and field observations, building on the framework developed by Duveiller et al.[85] for evaluating the influence of vegetation cover on surface energy exchange. Additional improvements of ESMs to better represent forest mortality, convection, and precipitation in tropical climate simulations also would help us more accurately assess the carbon costs of deforestation and evaluate their role in the changing climate system.

Our study provides a means to estimate the additional carbon losses associated with the regional to continental-scale biophysical effects of deforestation and their impact on regional climate and the carbon stocks of nearby undisturbed forests. For the Amazon, avoiding deforestation provides an additional $5.1 \pm 3.7\%$ benefit for aboveground vegetation carbon storage based on the model-simulated deforestation–climate effects and climate–vegetation carbon relationships derived from observations. For the Congo, this additionality is $3.8 \pm 2.5\%$. We find that such biophysical carbon costs of deforestation mainly arise from regional declines in precipitation and are further amplified by increases in surface air temperature. This, in combination with the estimated strong deforestation-driven climate–vegetation carbon feedback, emphasizes the additional threat from regional water stress triggered by deforestation and the potential effectiveness of climate mitigation strategies that maintain or expand robust tropical forest ecosystems.

## Methods

**CMIP6 simulations.** Precipitation, surface air temperature, vegetation carbon (being converted to the carbon in the aboveground biomass, AGB, using an empirical factor, that is, 0.8 for forests and 0.4 for savannas, see below), and tree cover fraction from 8 available Earth system models (ESMs[57–64]) (Table 1) participating in the Land Use Model Intercomparison Project (LUMIP) with model participating in phase 6 of Coupled Model Intercomparison Project (CMIP6)[55] were used in this study. The idealized deforestation simulations (deforest–glob) from LUMIP assume that a total forest area of 20 million km$^{-2}$ was linearly removed from the top 30% of forested area across the globe in 50 years. After 50 years, deforestation activity stopped, and most models were run for another

30 years for the purpose of reaching a stable status. The idealized global deforestation experiments caused a significant decline in tropical tree cover fraction, with the multimodel mean tree cover fraction decreasing by $44.7 \pm 6.0\%$, $38.7 \pm 8.8\%$ and $31.2 \pm 8.9\%$ in Amazon, Congo, and islands in tropical Asia, respectively (Table 2).

The above-mentioned variables from the preindustrial control (piControl) simulations of the eight models that participated in the LUMIP deforestation experiments were used to provide a referenced climate background in the tropics. This is justified by the fact that the LUMIP deforestation simulations start from a boundary condition that is identical to that in the piControl simulations. The first realization (r1) was selected for all 8 models for both the deforest–globe and piControl simulations, except for CESM2, for which the second realization (r2) in the deforest–globe simulations were selected due to a shifted rainfall climatology in r1.

We also downloaded these variables of the above 8 ESMs participating in the Coupled Climate–Carbon Cycle Model Intercomparison Project (C4MIP)[71], to calculate the $CO_2$-driven climate–vegetation carbon feedback parameter (detailed approach is described in detail below). The C4MIP experiments contain simulations of the idealized 1% per year increasing $CO_2$ experiments (1pctCO2), with the capability of $CO_2$ separately influencing the radiation components (1pctCO2-rad) and the carbon cycle model components (1pctCO2-bgc). These factorial experiment designs enable the isolation of the climate–carbon cycle feedback parameter (that is, the sensitivity of a carbon pool to climate warming, $\gamma$[69], with a unit of $kg\,C\,m^{-2}\,{}^\circ C^{-1}$). $\gamma$ was computed by subtracting the land carbon storage in 1pctCO2-bgc simulations from the 1pctCO2 simulations and by dividing this term by the corresponding climate warming in the 1pctCO2 simulations. In C4MIP experiments, 1pctCO2, 1pctCO2-rad, and 1pctCO2-bgc were run for each model for 140 years under a transient $CO_2$ increasing at a rate of 1% per year. By the end of 140 years, the atmospheric $CO_2$ concentration quadruples to about 1120 ppm. We thus represented the $4 \times CO_2$ effects on climate and aboveground vegetation carbon by computing their difference between the last and first 20-year averages for each 140-yr simulation in 1pctCO2 and 1pctCO2-bgc simulations. All variables from the above experiments were remapped to the 1-degree grid using the bilinear interpolation method from Climate Data Operator (CDO)[86].

**Observations**. Contemporary observations of mean annual precipitation and surface air temperature, derived from the Tropical Rainfall Measuring Mission (TRMM)[87] and Climate Research Unit (CRU TS4.04, ref. [88]), were used to obtain the empirical relationship between climate and the AGB. The observational AGB (unit: $Mg\,ha^{-1}$, being converted to Mg C ha$^{-1}$ using a factor of 0.5, ref. [89]) was derived from the European Space Agency Climate Change Initiative (ESA–CCI) BIOMASS project[3]. ESA–CCI biomass map provides detailed information of the aboveground vegetation carbon storage during the year 2010, 2017, 2018 at a spatial resolution of $100 \times 100$ m. In this study, we used the AGB map during the year of 2017. The accuracy of ESA–CCI AGB has been improved when compared to the previous version (that is, AGB from GlobBiomass project, used in land surface model evaluation[89]), with the upper limit of AGB relative error being 20% where AGB exceeds 50 Mg ha$^{-1}$ and a fixed error of 10 Mg ha$^{-1}$ where the AGB is below that limit (see Product Validation & Algorithm Selection Report Version 2 in https://climate.esa.int/en/projects/biomass/key-documents/). Nevertheless, ESA–CCI may still underestimate AGB in wet tropics because both L– and C–band backscatter data saturate at high AGB levels when AGB values keep increasing. We averaged the original AGB data to a $1° \times 1°$ grid, corresponding to the unified resolution of the LUMIP model output used in this study. Accordingly, observed satellite precipitation from TRMM and air temperature from CRU in the year of 2017 were downloaded and aggregated to the same 1-degree resolution.

Other observational data include AGB data from ref. [1] and ref. [2], land cover from the Moderate Resolution Imaging Spectroradiometer (MODIS) data (MCD 12C1, distinguishing the land fraction within each 1-degree pixel), and MODIS vegetation continuous fields (VCF) data (MOD44B, used as the observational tree cover fraction as shown in Supplementary Fig. 7) during the year of 2017. We also used the observed precipitation from Global Precipitation Climatology Centre (GPCC)[90] and estimated climatological (1987–2016 average) precipitation in the Amazon, Congo, and tropical Asia (Supplementary Table 4).

**Deforestation effects on rainfall and temperature**. For each model, the deforestation impacts on mean annual precipitation and temperature were computed as the difference of the last 30-year average between the deforest–glob and piControl simulations. For the multimodel mean, we calculated the agreement of these eight models on the sign of the deforestation-caused change in precipitation and temperature, with at least six out of eight models agreement indicated by the dotted area shown in Fig. 1. A recent study by Boysen et al.[65] has given an overview of the simulated deforestation effects on global climate. Although tropical rainfall and air temperature may be perturbed significantly by deforestation in the extratropical regions (in particular, see remote effects by high-latitude deforestation in Devaraju et al.[26]), most models agree that deforestation causes an averaged warming and decline in rainfall due to the large-scale decline in evapotranspiration within the tropics[65].

To confirm that most of the tropical climate response originates from tropical deforestation in the LUMIP experiments, we conducted an additional experiment

with CESM2. This experiment exactly followed the LUMIP protocol[55] but excluded deforestation in the extratropics poleward of 23°S–23°N. We ran the model for 80 years, with tropical tree cover losses exactly equivalent to those in CESM2 participating in the LUMIP deforest–glob experiments. Different from the LUMIP deforest–glob simulations, in this experiment (CESM–trop) all tree cover fraction was invariant in the extratropics. To eliminate the influence from model version or initial conditions, we ran the same model in a configuration of the preindustrial conditions for another 30 years (CESM2–ctl). The difference in precipitation and temperature over the last 30 years between CESM2–trop and CESM2–ctl thus represents the climate effects of deforestation only in the tropics in CESM2. This experiment confirmed that most of the simulated changes in precipitation and temperature over tropical forests originated from deforestation within the tropics. Nevertheless, more work is needed to systematically examine tropical and extratropical deforestation contributions (e.g., remote effects on tropical monsoon precipitation by high-latitude deforestation via shifting the intertropical convergence zone[26]) to regional climate across the full suite of CMIP models.

**Spatial AGB sensitivity to rainfall and air temperature**. To obtain the climate sensitivity of the aboveground biomass (AGB), we first applied a multiple linear regression model to the observational or simulated grids of biomass as a function of the mean annual precipitation (MAP) and temperature (MAT) across the tropics (23°S–23°N) (Table 3).

$$AGB = a * MAP + b * MAT + \varepsilon \tag{1}$$

The regression was applied to all land grids in the observational datasets (see above descriptions) for the year of 2017 within 23°S–23°N, excluding those with a MAP lower than 100 mm yr$^{-1}$ and land fraction lower than 0.5 (mainly the edge pixels on islands in tropical Asia). The land fraction was computed using MCD12C1 land cover. The coefficients of a and b were shown in Table 3 for both the observations and CMIP6 mean.

Despite using all land grid cells in the regression method, we found that the AGB spatial sensitivity varies as a function of climatological mean rainfall (Fig. 2c). We thus applied the regression method to estimate the AGB–climate sensitivity at different precipitation levels. Each level of precipitation (that is, from 600 to 3100 mm yr$^{-1}$ at an interval of 100 mm yr$^{-1}$) is the center rainfall condition of each moving window spanning ±500 mm yr$^{-1}$ (for example, land grid cells with precipitation from 100 mm yr$^{-1}$ to 1100 mm yr$^{-1}$ were used in the regression for estimating AGB–climate sensitivity at the precipitation level of 600 mm yr$^{-1}$) (Fig. 2c). This approach was also used to estimate the subsequent AGB costs of tropical deforestation impacts through changing the regional climate.

Observational climate sensitivity of the AGB may be influenced by spatial variation in contemporary disturbance regimes, including fire and agriculture, which were not considered in the piControl simulations of the models. To test the robustness of the regression-derived parameters in the observations, we applied a similar multiple linear regression model to the simulated AGB and precipitation, and temperature from the piControl simulations of LUMIP models. On basis of previously identified empirical ratios for aboveground to total biomass (that is, 0.8 for forests and 0.4 for savannas[91]), LUMIP ESMs simulated total vegetation carbon was multiplied by a mean factor weighted by the simulated tree cover fraction (that is, mean factor $= 0.8 \times$ tree cover $+ 0.4\times$ (1−tree cover)) and converted to AGB carbon.

An implicit assumption here is that aboveground vegetation carbon in the tropics is influenced by the precipitation- and temperature-induced changes in environmental factors (including the subsequent changes in soil moisture and the vapor pressure deficit, VPD), which influences aboveground vegetation carbon through changes in vegetation physiological processes (for example, stomatal closure). We also computed the observed VPD, mean annual maximum temperature (MAXT), the seasonality of precipitation ($P_{amp}$, defined as the difference in rainfall between the month with the maximum value and the month with the minimum value), the seasonality of temperature ($T_{amp}$, defined as the difference in temperature between the month with the maximum value and the month with the minimum value), and precipitation in the driest quarter (PRD, defined as the minimum rainfall of consecutive 3 months throughout the year). We diagnosed the relationship of these metrics with MAT and MAP using the Belsley collinearity diagnostics using the software MATLAB. We found that $T_{amp}$, MAXT, and VPD have high spatial collinearity with the MAT, while $P_{amp}$ and PRD have high spatial collinearity with the MAP. Models with these extra variables did not significantly improve the goodness of fit of the regression model (Supplementary Tables 2 and 3). This suggests that these water and heat stress factors that are critical for plant physiological processes covary spatially to a high degree with the mean annual precipitation and surface air temperature.

**Biophysical AGB loss from tropical deforestation**. We estimated the biophysical AGB costs of tropical deforestation (Fig. 3a–c) by multiplying the deforestation-induced climate change from LUMIP (Fig. 1) and the observed AGB carbon sensitivity to mean annual precipitation and air temperature (Fig. 2c). We focused on three tropical continents (Amazon, Congo, and islands in tropical Asia) with a high fraction of intact forests where the signal of deforestation–climate impacts is also relatively robust (Fig. 1). Instead of using a unified AGB sensitivity to MAP and MAT (shown in Table 3), we used the moving window AGB–climate

sensitivity at different rainfall levels to estimate the carbon costs of tropical deforestation-driven climate change as the AGB–climate sensitivity varies with the rainfall background (Fig. 2c). When applying this moving window approach, we inferred the AGB–climate sensitivity at each grid cell by its background rainfall level, which was computed as the GPCC[90] climatological precipitation average plus the deforestation-induced change in relative rainfall (%) from the LUMIP experiments. To measure the relative role of these biophysically driven carbon losses, we first estimated the direct AGB losses due to losses of the multimodel mean tree cover in LUMIP. The LUMIP ESMs simulated AGB loss along with deforestation cannot represent the realistic values as the ESMs have a negative bias for AGB in tropical regions with high fraction of forests (Supplementary Fig. 4). To estimate the realistic direct AGB loss due to deforestation, we used the observational relationship between AGB and tree cover fraction, both of which were derived from the satellite observations. Supplementary Fig. 7 shows that a 10% loss in tree cover fraction corresponds to 19–22 Mg C ha$^{-1}$ loss in AGB for the observations. Using this ratio in three tropical regions, combined with the multimodel mean tree cover loss, we estimated that the tree cover losses by 44.7 ± 6.0%, 38.7 ± 8.8%, and 31.2 ± 8.9% in LUMIP corresponds to direct AGB carbon losses by −98.3 ± 13.2 Mg C ha$^{-1}$, −75.5 ± 16.7 Mg C ha$^{-1}$, and −62.4 ± 17.8 Mg C ha$^{-1}$ in the Amazon, Congo, and maritime continent in tropical Asia, respectively (Table 2). The relative role of biophysical carbon costs of tropical deforestation was calculated as the ratio of the biophysically driven AGB loss to the estimated direct AGB losses (Table 2 and Fig. 3d–f).

By assuming the ESM-diagnosed deforestation–climate relationships are a linear function of the deforestation level[92], we estimated the cumulative impacts of past deforestation on regional climate and carbon stocks. To quantify the past history of deforestation, we used the Land Use Harmonization (LUHv2h) dataset[68] spanning the period from 1850 to 2015. Considering the Amazon basin, for example, (also see Supplementary Table 4), the mean annual precipitation decline caused by primary forest loss (−11.5%) was estimated to be −1.7%, and the mean annual warming was estimated to be 0.13 °C. Applying the climate sensitivity of AGB (Fig. 2c), this yielded a cumulative AGB loss of 1.3 Mg C ha$^{-1}$ from rainfall decline and an additional gain of 0.04 Mg C ha$^{-1}$ from warming. Together, these add up to about 741 Tg C for the Amazon basin (area, 5,840,000 km$^2$). Using the same approach (Supplementary Table 4), we estimated the biophysically driven AGB loss for the Congo to be 200 Tg C 1850 to 2015 from losses in forest cover (Supplementary Table 4).

**CO$_2$- and deforestation-driven climate–vegetation carbon feedback.** The CO$_2$-driven climate–carbon cycle feedback considers the isolation of the land carbon–climate sensitivity, γ, from fully coupled (1pctCO2) and biogeochemically coupled (1pctCO2-bgc) idealized climate simulations[56,69,70]. Here we modified this approach to consider only the contribution to γ arising from changes in AGB at each grid cell (CMIP6 simulated vegetation carbon was converted to the AGB using an empirical factor, that is, 0.8 for forests and 0.4 for savannas):

$$\gamma_{AGB}^{CO2} = \frac{\triangle C'_{AGB} - \triangle C^*_{AGB}}{\triangle T'} \quad (2)$$

where $\triangle C'_{AGB}$, and $\triangle C^*_{AGB}$ represent the changes in the AGB from 1pctCO2 and 1pctCO2-bgc simulations, and $\triangle T'$ denotes the increases in surface air temperature from 1pctCO2. The changes of AGB carbon and air temperature from these two simulations were quantified as the difference between the last and first 20-year average (for both two simulations, 140 years in total). $\gamma_{AGB}^{CO2}$ thus represents the CO$_2$-driven climate–carbon feedback parameter (see the first row in Fig. 5).

The CO$_2$-driven climate–carbon feedback framework considers the biogeochemical climate effects of increased CO$_2$ concentration that may come from both fossil fuel and land-use change carbon emissions but does not consider the contribution from the biophysical effect of tropical deforestation, and therefore may underestimate the land-use risk for the carbon–climate feedback in the tropics. Using the LUMIP simulated biophysical deforestation effects and the sensitivity of the AGB to climate derived from the observations, we calculated the deforestation-driven carbon–climate feedback parameter as follows:

$$\gamma_{AGB}^{def.biophys} = \frac{\triangle C_{AGB}^{def.biophys}}{\triangle T^{def.biophys}} \quad (3)$$

where $\gamma_{AGB}^{def.biophys}$ denote the climate sensitivity of the AGB carbon under tropical deforestation. $\triangle C_{AGB}^{def.biophys}$ and $\triangle T^{def.biophys}$ represent the biophysical effects of deforestation on tropical AGB carbon and air temperature, respectively. $\triangle T^{def.biophys}$ was computed as the LUMIP models simulated air temperature change in the tropics, while $\triangle C_{AGB}^{def.biophys}$ was calculated as the sum of the deforestation-caused changes in mean annual precipitation and temperature, multiplied by the observational AGB sensitivity to precipitation and temperature, respectively. $\gamma_{AGB}^{CO2}$ and $\gamma_{AGB}^{def.biophys}$ are shown in the first and second row of Fig. 5, respectively.

## Data availability

All CMIP6 simulations are publicly available via https://esgf-node.llnl.gov/projects/cmip6/. Observational precipitation from TRMM 3B43 and climate variables from CRU TS4.04 are accessible via the websites: https://disc.gsfc.nasa.gov/datasets/TRMM_3B43_7/ summary/ and https://crudata.uea.ac.uk/cru/data/hrg/cru_ts_4.04/, respectively. ESA–CCI AGB is available via: https://climate.esa.int/en/projects/biomass/. AGB from ref. [1] is available via: https://www.ilamb.org/ILAMB-Data/DATA/biomass/Tropical/. AGB from ref. [2] is available via: https://developers.google.com/earth-engine/datasets/catalog/WHRC_biomass_tropical. MODIS MCD12C1 and MOD44B are available via EarthData: https://earthdata.nasa.gov/. Processed data for this study have been deposited in Li Yue: (2022). Data supporting figures of deforestation_carbon_biophysics (v1.0). Zenodo. https://doi.org/10.5281/zenodo.6326365.

## Code availability

Code and scripts of CESM2 can be downloaded through the website via: https://escomp.github.io/CESM/versions/cesm2.1/html/downloading_cesm.html. Scripts for all figures and tables are at https://github.com/YueLi92/deforestation_carbon_biophysics, which has been deposited in Li, Yue: (2022). Code supporting deforestation_carbon_biophysics (v1.0). Zenodo. https://doi.org/10.5281/zenodo.6323730.

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

## Acknowledgements

This work was funded by NASA's SERVIR Amazonia program in the form of a grant (80NSSC20K0590) to D.C.M., P.M.B., and J.T.R., by NASA's Carbon Monitoring System (CMS) and Modeling, Analysis and Prediction (MAP) programs (80NSSC18K0179), and by the US. Dept. of Energy Office of Science RUBISCO Science Focus Area. D.M.L. is supported by the National Center for Atmospheric Research, which is a major facility sponsored by the NSF under Cooperative Agreement No. 1852977.

## Author contributions

Y.L. and J.T.R. designed the research; Y.L. performed data analysis and model simulations; Y.L. and J.T.R. drafted the manuscript, with discussions and contributions from P.M.B., D.C.M., D.M.L., and H.Y.; All authors reviewed and revised the manuscript.

## Competing interests

The authors declare no competing interests.
