## [Peer Review File · Nature Communications]

Deforestation-induced climate change reduces carbon storage in remaining tropical forestsREVIEWER COMMENTS

Reviewer #1 (Remarks to the Author):

Review of “Deforestation-induced climate change reduces carbon storage in remaining tropical forests” by Yue Li and others

Deforestation has both biogeochemical and biophysical effects. In the tropics, the biophysical effect of deforestation is an increase in temperature and decrease in precipitation. These two changes have the potential to further reduce the biomass in the tropical forests. This paper quantifies the amount of reduction in aboveground biomass in the tropical forests due to these biophysical effects. The authors find that, by inference, that the biomass in Amazon is likely now less by 740 TgC because of the warming and reduction in rainfall caused by the biophysical effects deforestation during the historical period.

This is certainly an interesting quantification of the “feedback” from the biophysical effects. However, aren’t biogeochemical and biophysical effects, by definition, “feedbacks” in the climate system. Positive feedbacks run in circular loops with damping from negative feedbacks. From this point of view, the authors have investigated a second order effect from the biophysical effects of deforestation. The first order effect on climate is the change in the warming and decrease in precipitation. Second order effects are usually small as demonstrated by the authors – a small 6.8 % additional loss of above ground biomass in the Amazon. Indeed, the additional loss of AGB is only ~ 1 TgC for the entire historical period. This is too small compared to the amount of cumulative deforestation flux during the whole historical period or the amount of carbon stocks in tropical forests (~200 PgC).

I appreciate that the work is technically rigorous and sound. The quantification is original and novel. The presentation is good though very technical. This paper is certainly suitable for publication in more technical journals on bio geosciences. Therefore, my view is that this paper may not be suitable for publication in Natural Communications which should be accessible and appreciated by a wider audience. I am also concerned that the authors are giving too much importance to a small secondary effect (0.5 % effect).

Specific comments:

1. Line 23: “moister climate” should be “moist climate”
2. Line 37: The flux values for the period 2010-2019 should be also specified.
3. Line 40: Does deforestation here refer to the global or tropical deforestation. This should be clarified in this sentence.
1. Lines 44-56: It is strange that some of the most important and seminal papers on the biophysical impacts of deforestation are not referred in this paper at all. The notable omissions are Bonan et al., 1992 Nature, Betts 2000 Nature, Betts et al. 2007; Gibbard et al. GRL 2005, Bala et al. 2007 PNAS, Bathiany et al. 2010 BioGeoSci, Devaraju et al. 2015 PNAS, and Devaraju et al. 2015 PCE. These references should be cited in the discussions of biophysical effects of deforestation.
4. Lines 62-64: The spatial extent of deforestation is also a contributing factor.
5. Line 133: “smaller decrease in cloud cover”. This can certainly be checked in the LUMIP simulations. A figure on cloud fraction change in the supplemental file would be useful to the readers to understand the weaker response in Congo.
6. Line 293: How do physiological effect cause an increase in precipitation? I believe physiological effects cause a decrease in evapotranspiration. Do we know for sure it also causes an increase in precipitation and why? This issue should be discussed here.
7. Lines 302-303: I believe that the parameter γ does not care whether the CO₂ emissions are the result of fossil fuel burning or land use change. I agree that it does not include biophysical effect of land use change. This should be distinguished and clarified.
8. Line 354: 20 million km². What fraction of the total forest cover is removed in this experiment? This fraction should be also mentioned to understand the magnitude of deforestation.

9. Line 395: Why is the error smaller when the AGB is larger? This may be briefly discussed.
10. Line 398: Why “underestimate”?
11. Lines 417-420 and lines 431-435: This is contrary to what the paper Devaraju et al. 2015 PNAS finds. It is shown in this PNAS paper that remote high-latitude deforestation has larger effect on tropical precipitation than local deforestation.
12. Line 494: It is shown that the AGB climate sensitivity varies with the rainfall background. Does it also vary with temperature background? This question should be briefly addressed here.
13. Lines 536-538: I believe that the climate carbon feedback does not care whether the CO₂ emissions are the result of fossil fuel burning or land use change. I agree that it does not include biophysical effect of land use change. This should be distinguished and clarified.
14. Table 2: Absolute changes in precipitation and AGB are difficult to understand and appreciate and different regions may have different base values. Therefore, I suggest the authors to also list the % changes in these quantities in the table.
15. Table 2: the last 3 columns should clearly indicate that the quantities shown are AGB and not total forest carbon (which includes soil carbon)
16. Figure 1, top three panels: As indicated earlier, it would be useful to understand and appreciate if the % changes in precipitation are shown here or in a supplemental figure.
17. Figure 2, bottom 2 panels: Sensitivity is estimated for various base precipitation values. Why not also estimate the sensitivity for various background T values?
18. Figure 3, top 3 panels: It would be useful to understand and appreciate if the % changes in AGB are also shown here or in a supplemental figure.
19. Figure 4, top 3 panels: It would be useful to understand and appreciate if the % changes in AGB are also shown here or in a supplemental figure.
20. Figure S2: A panel that shows the differences between top and bottom panels should be also provided for understanding the differences between simulations and observations.
21. Table S4, footnote: More parentheses should be used to clearly indicate which terms are in the denominator and which ones are in the numerator.

Reviewer #2 (Remarks to the Author):

This study investigates the impacts of deforestation-induced climate change (precipitation and temperature) on carbon storage in remaining tropical forests using Earth system model simulations and satellite-derived aboveground biomass datasets. Previous relevant studies mainly analyze the losses of aboveground biomass directly from deforestation. This is a very interesting and important study to improve forest aboveground biomass loss under deforestation. I have some comments, which may help authors improve this manuscript.

Major comments:

I do not have much knowledge about these Earth system models. In this study, the model simulations have been used in estimating aboveground biomass losses from deforestation and deforestation-induced climate change, and precipitation and temperature changes caused by deforestation. What are the uncertainties of these aboveground biomass losses, and precipitation and temperature changes estimated by Earth System models, comparing to the long-term satellite observations or the field observations?

Minor comments:

Line 44-46: Lee et al paper is not about the tropical forest and may not be cited to support this sentence.

Line 98: The definition of carbon cost can be moved from the Results section (Line 204) to here.

Line 101: What is the meaning of gamma? It is not clear to readers.

Line 142: When analyzing the sensitivity of tropical aboveground biomass to climate, why did you use a moving window (± 500 mm) for precipitation?

Line 172: The land cover map may be needed in supplementary materials to help readers understand the tropics' spatial distribution of forest and savanna.

We would like to thank the two reviewers for their valuable and constructive suggestions on improving our manuscript, “Deforestation-induced climate change reduces carbon storage in remaining tropical forests.” Below, we provide a detailed point-by-point response to the reviewers’ comments. Figure and table numbers cited in this letter correspond to the revised manuscript (e.g., Fig. 2, Table 4), supplementary materials (e.g., Supplementary Fig. 4), and reviewer response letter (e.g., Fig. R1). The numbers of page and lines (e.g., Page: 5, Lines: 84-90) correspond to the version of the manuscript without markup.

Important changes to the manuscript include new figures showing the influence of deforestation on precipitation as a relative change (in percent) (Supplementary Fig. 2) and cloud cover (Supplementary Fig. 3), a new panel in Supplementary Fig. 4 showing the difference between model and observed aboveground biomass, a new figure showing the distribution of land cover (Supplementary Fig. 5) and a new table in the main text (Table 4) highlighting the importance for forest offset projects of integrating the biophysical climate effects of deforestation on vegetation carbon stocks with carbon losses from edge effects. We also added more discussion of the uncertainties associated with Earth system model simulations of biomass loss, temperature, and precipitation changes caused by deforestation.

We also performed minor editing of the manuscript to improve clarity. In revising our paper, when computing the biophysical impacts of deforestation on remaining aboveground biomass (AGB), we used observed maps of baseline precipitation rather than model estimates in the earlier draft. We did this to be more consistent with all available observational constraints when applying the biomass–climate relationships from Fig. 2. In the Amazon, for example, our estimate of the biophysical impacts of deforestation on intact forest carbon decreased from 6.8% to 5.1%. This improved overall consistency of projected changes shown in the figures (Fig. 3 for example). We believe our manuscript is one of the first to quantify the magnitude of these changes and that they are still very important from a natural climate solutions perspective.

Reviewers’ comments:

Reviewer #1 (Remarks to the Author):

Review of “Deforestation-induced climate change reduces carbon storage in remaining tropical forests” by Yue Li and others.

Deforestation has both biogeochemical and biophysical effects. In the tropics, the biophysical effect of deforestation is an increase in temperature and decrease in precipitation. These two changes have the potential to further reduce the biomass in the tropical forests. This paper quantifies the amount of reduction in aboveground biomass in the tropical forests due to these biophysical effects. The authors find that, by inference, that the biomass in Amazon is likely now less by 740 TgC because of the warming and reduction in rainfall caused by the biophysical effects deforestation during the historical period.

Response: We thank the reviewer for the rigorous assessment on our work. We have carefully revised the manuscript and provided summaries of recent advances to highlight the importance of

considering the biophysical feedback of tropical deforestation on the aboveground biomass in the current carbon accounting system. Point-by-point responses are offered below.

This is certainly an interesting quantification of the “feedback” from the biophysical effects. However, aren’t biogeochemical and biophysical effects, by definition, “feedbacks” in the climate system. Positive feedbacks run in circular loops with damping from negative feedbacks. From this point of view, the authors have investigated a second order effect from the biophysical effects of deforestation. The first order effect on climate is the change in the warming and decrease in precipitation. Second order effects are usually small as demonstrated by the authors – a small 6.8 % additional loss of above ground biomass in the Amazon. Indeed, the additional loss of AGB is only ~ 1 TgC for the entire historical period. This is too small compared to the amount of cumulative deforestation flux during the whole historical period or the amount of carbon stocks in tropical forests (~200 PgC).

I appreciate that the work is technically rigorous and sound. The quantification is original and novel. The presentation is good though very technical. This paper is certainly suitable for publication in more technical journals on bio geosciences. Therefore, my view is that this paper may not be suitable for publication in Natural Communications which should be accessible and appreciated by a wider audience. I am also concerned that the authors are giving too much importance to a small secondary effect (0.5 % effect).

Response: We appreciate the reviewer’s recognition of the originality and novelty of our quantification of biophysical feedback to tropical vegetation carbon storage. Indeed, the positive feedback from warming and drying of tropical deforestation has been found in this study to amplify the loss in the aboveground vegetation carbon storage. We acknowledge that the additional loss of the aboveground biomass (AGB) in the Amazon is small, but we would argue this is not negligible in the context of climate mitigation policy. In particular, we believe this biophysical feedback should be integrated in the forest carbon offset programs as there is a growing interest in implementing avoided deforestation or reforestation as natural climate solutions. The interest in these strategies is why we think our paper is suitable for the more general science audience of *Nature Communications*. We provide more information below and modified the text in the manuscript in several places, including addition of a new table of listing the additional indirect carbon benefits from avoiding deforestation in the tropics, to address the reviewer’s concern about the importance of our study. We also believe the reviewer’s assertion that the biophysical climate feedback is at a 0.5% level is incorrect, and we explain why below.

To address the reviewer’s concern about the importance of our study and the magnitude of the effect, we have modified in the introduction and discussion. Specific modifications include the following:

(Page: 5, Lines: 84-90, in the introduction) We added the following text in the introduction to further motivate our study from a natural climate solutions perspective:

“To determine the magnitude and sign of these larger-scale interactions, here we quantify the influence of deforestation-driven climate change on the carbon storage of forests across different tropical continents. This is important because natural climate solutions are gaining attention as a

possible mechanism to slow climate warming. In forest carbon offset programs, a critical need is to provide an accurate estimate of the carbon and climate benefits of a land management action (e.g., avoided deforestation), thus enabling more effective valuation of the carbon credits issued for a specific project.”

(Page: 14-15, Lines: 290-296, in the discussion) “An important next step in this context is to combine the deforestation biophysical climate effect identified here with AGB losses associated with local edge effects^{54,77-79} in order to estimate an integrated indirect carbon benefit associated with avoided deforestation (or reforestation) projects in the tropics. A preliminary comparison of these two mechanisms and their total effects are provided in Table 4. In the Amazon, these indirect benefits of avoided deforestation may sum to be 41% (with a range of 26-65%) higher than carbon contained within the project boundaries.”

Table 4. Additional benefits for avoiding deforestation or reforestation associated with the indirect local and regional effects of deforestation on aboveground biomass (AGB).

Mechanisms	Impact of edge or climate feedback on AGB		Citation
	Mean	Range	
Local edge effects (from changes in microclimate and fire)	36% (Amazon)*	25-56%	Junior et al. ⁵⁴ Zhao et al. ⁷⁷ Ordway and Asner ⁷⁸
	19% (Congo) [§]	18-20%	
	10% (Tropical Asia) [#]	7-13%	
Regional climate feedback (from changes in rainfall and temperature)	5.1% (Amazon)	1.4-8.8%	This study
	3.8% (Congo)	1.3-6.3%	
	0.5% (Tropical Asia)	0.0-3.7%	
Local edge + regional climate feedback	41% (Amazon)	26-65%	
	23% (Congo)	19-26%	
	11% (Tropical Asia)	7-17%	

*Ratio between total gross carbon loss from edge effect and that from deforestation during 2001-2015.

[§]Ratio between carbon loss from edge effect and that from deforestation for scenarios 1-3 under the Representative Concentration Pathway 8.5 (RCP8.5) for the entire continent of Africa.

[#]Ratio of biomass loss has been reported to range from 16% to 30% according to forest sites observations in Sabah, Malaysian Borneo⁷⁸. This ratio has been multiplied by an average fraction of forest fragmentation of 31% (Fischer et al.⁷⁹), and further divided by the remaining forest fraction of 69% that could experience potential large-scale deforestation.

We also wish to note that we highlight the importance of the biophysical feedback from deforestation in the tropics also because these additional vegetation carbon losses have the potential to develop a positive climate feedback with respect to tropical ecosystems reaching their tipping point through a series of mechanisms as a result of drying and warming such as increasing the length of dry season, favoring more frequent extreme drought events and causing a higher likelihood of fires. It is currently a challenge to identify where and when tropical forest ecosystems may reach their climate tipping point due to deforestation and forest cover loss but reporting the carbon consequences of these biophysical climate feedback may contribute to a broader

understanding of the magnitude of the positive climate–tropical forest feedback loop. We have stated the importance of reporting the aboveground carbon loss of biophysical feedback of deforestation in climate tipping point in the previous version of the manuscript:

(Page 5, Lines: 75-78, in the introduction) “Warming and drying also promotes drought and wildfires^{15,48}, which greatly increases the risk of regional forest dieback and the associated loss of the aboveground biomass⁴⁹. An ensuing climate tipping point, once triggered, may cause local ecosystems to move toward an alternate stable state^{5,32,50,51}”

(Page 16, Lines: 319-326, in the discussion) “Nevertheless, the change of mean climate by deforestation (that is, less rainfall and warming), as revealed in our analysis of the LUMIP–CMIP6 simulations, still implies the increasing possibility of a lengthening dry season⁸², an increasing amplitude of extreme drought events⁸³, and a higher likelihood of wildfires^{5,10,15} in the tropics if deforestation continues in the future. More broadly, the high levels of uncertainty regarding how radiative, physiological, and land cover change mechanisms influence tropical precipitation makes it challenging to accurately predict climate tipping points in the tropics.”

Finally, we would like to point out that we do not think it is correct to compare the additional loss of AGB of about 741 Tg C to the cumulative deforestation flux of about 200 Pg C, which is a number quoted from Erb et al. (2018) and an estimation for the pan tropical AGB loss for a much longer timespan. The 741 Tg C estimate we report in the discussion as a consequence of deforestation-driven regional climate change should be compared to a direct deforestation loss in the Amazon of about 14.6 Pg C. We derived this estimate using forest cover change estimates from the Land Use Harmonization (LUHv2h) dataset (Hurtt et al. 2020) and the observed AGB relationship with forest cover (Supplementary Fig. 7). Thus, our estimate of historical AGB losses from the regional climate change effect of deforestation is about 5% for the Amazon. This estimate is consistent with the number reported in the results section ($5.1 \pm 3.7\%$, Table 2).

(Page: 2, Lines: 20-22, in the abstract) “In the Amazon, warming and drying arising from deforestation results in an additional $5.1 \pm 3.7\%$ loss of aboveground biomass. Biophysical effects also amplify carbon losses in the Congo ($3.8 \pm 2.5\%$)”

(Page: 12, Lines: 228-236, in the results) “Fig. 3 shows the spatial pattern of the biophysical effects of deforestation on AGB and their percent contribution to total biomass loss related to the direct effects of deforestation. Despite the widespread decline in rainfall and warming, the largest AGB loss due to biophysical feedback occurs in the eastern Amazon where the additional AGB loss is as high as 14 Mg C ha^{-1} (17%) (Fig. 3a, d). There is no additional AGB loss predicted for the northwestern Amazon due to its high baseline precipitation level that reduces the sensitivity of AGB to changes in precipitation or temperature (Fig. 2c). Biophysical effects also amplify the AGB loss in central Congo Basin by up to 9 Mg C ha^{-1} (11%) but do not lead to any additional AGB loss in tropical Asia as a consequence of its high precipitation baseline (Fig. 3b–c, e–f).”

References

Erb, K. H. et al. Unexpectedly large impact of forest management and grazing on global vegetation biomass. *Nature* **553**, 73-76 (2018).

Fischer, R. et al. Accelerated forest fragmentation leads to critical increase in tropical forest edge area. *Sci. Adv.* **7**, eabg7012 (2021).

Hurt, G. C. et al. Harmonization of global land use change and management for the period 850–2100 (LUH2) for CMIP6. *Geosci. Model Dev.* **13**, 5425–5464 (2020).

Junior, C. H. S. et al. Persistent collapse of biomass in Amazonian forest edges following deforestation leads to unaccounted carbon losses. *Sci. Adv.* **6**, eaaz8360 (2020).

Ordway, E. M., & Asner, G. P. Carbon declines along tropical forest edges correspond to heterogeneous effects on canopy structure and function. *Proc. Natl. Acad. Sci. U.S.A.* **117**, 7863–7870 (2020).

Zhao, Z. et al. Fire enhances forest degradation within forest edge zones in Africa. *Nat. Geosci.* **14**, 479–483 (2021).

Specific comments:

1. Line 23: “*moister climate*” should be “*moist climate*”

Response: We have changed it into “high levels of annual mean precipitation” in the revised manuscript (Page: 2, Line: 23).

2. Line 37: *The flux values for the period 2010-2019 should be also specified.*

Response: We agree with the reviewer that specifying the flux values for the period 2010-2019 would be helpful to understand the declined land-use carbon emission since the 2000s due to declined tropical deforestation. We added the following text to address this point:

(Page: 3, Lines: 36-39) “Since the 2000s, rates of tropical deforestation have slowed, contributing to an overall decline in the global carbon flux from land use change (e.g., from 1.9 Pg C yr⁻¹ in 1997 to 1.0 Pg C yr⁻¹ during 2010–2019, ref. ^{11,13}).”

3. Line 40: *Does deforestation here refer to the global or tropical deforestation. This should be clarified in this sentence.*

Response: The “deforestation” here refers to “tropical deforestation” and we have modified the text in the revised manuscript.

(Page: 3, Lines: 41-43) “Altogether, cumulative carbon emissions from tropical deforestation and other land–cover changes in the tropics over the past several centuries are comparable to the current aboveground vegetation carbon stock¹⁷.”

1. Lines 44-56: *It is strange that some of the most important and seminal papers on the biophysical impacts of deforestation are not referred in this paper at all. The notable omissions are Bonan et al., 1992 Nature, Betts 2000 Nature, Betts et al. 2007; Gibbard et al. GRL 2005, Bala et al. 2007*

PNAS, Bathiany et al. 2010 BioGeoSci, Devaraju et al. 2015 PNAS, and Devaraju et al. 2015 PCE. These references should be cited in the discussions of biophysical effects of deforestation.

Response: We thank the reviewer for the constructive suggestions to these great papers, which all use a single model to simulate the biophysical impacts of deforestation. Our results of the multi-model analysis are consistent with these previous studies. Since our study focuses on the tropics, we have cited the references that the reviewer identified that have a scope covering tropical ecosystems (including Betts et al. 2007; Gibbard et al. 2005; Bala et al. 2007; Bathiany et al. 2010; Devaraju et al. 2015a; Devaraju et al. 2015b) both in the introduction and discussion of the revised manuscript.

(Page: 3, Lines: 47-49, in the introduction) “The influence of these biophysical changes has been long appreciated by regional and global climate modeling communities²²⁻²⁷ and are known to contribute to regional warming and drying²⁸⁻³²,”

(Page: 15, Lines: 298-301, in the discussion) “The positive biophysical climate feedback of tropical deforestation (e.g., warming and reduced precipitation as revealed by previous single model-based studies²²⁻²⁷ and our multi-model analysis) has important implications for assessing future climate risks of tropical moist forests^{50,80}.”

(Page: 32-33, Lines: 653-667)

“22. Betts, R. Implications of land ecosystem-atmosphere interactions for strategies for climate change adaptation and mitigation. *Tellus Ser. B-Chem. Phys. Meteorol.* **59**, 602-615 (2007).

23. Gibbard, S., Caldeira, K., Bala, G., Phillips, T. J., & Wickett, M. Climate effects of global land cover change. *Geophys. Res. Lett.* **32**, L23705 (2005).

24. Bala, G. et al. Combined climate and carbon-cycle effects of large-scale deforestation. *Proc. Natl. Acad. Sci. U.S.A.* **104**, 6550-6555 (2007).

25. Bathiany, S., Claussen, M., Brovkin, V., Raddatz, T., & Gayler, V. Combined biogeophysical and biogeochemical effects of large-scale forest cover changes in the MPI earth system model. *Biogeosciences.* **7**, 1383-1399 (2010).

26. Devaraju, N., Bala, G., & Modak, A. Effects of large-scale deforestation on precipitation in the monsoon regions: Remote versus local effects. *Proc. Natl. Acad. Sci. U.S.A.* **112**, 3257-3262 (2015).

27. Devaraju, N., Bala, G., & Nemani, R. Modelling the influence of land-use changes on biophysical and biochemical interactions at regional and global scales. *Plant Cell Environ.* **38**, 1931-1946 (2015).”

4. Lines 62-64: *The spatial extent of deforestation is also a contributing factor.*

Response: We thank the reviewer for the rigorous consideration and have added this contributing factor in the introduction part of the revised manuscript.

(Page: 4, Lines: 61-65) “The regional climate response to deforestation in tropical Africa and Southeast Asia is weaker in magnitude, likely as consequence of different forms of land cover change, different climate baseline states, spatial patterns of deforestation, and geographical differences in topography and proximity of forests to nearby ocean regions⁴⁴”

5. Line 133: “smaller decrease in cloud cover”. This can certainly be checked in the LUMIP simulations. A figure on cloud fraction change in the supplemental file would be useful to the readers to understand the weaker response in Congo.

Response: We thank the reviewer for the constructive suggestion. We have performed additional analysis by investigating the cloud cover response to deforestation in the LUMIP simulations. A figure on the cloud fraction changes is shown below and has been added into the supplemental materials to illustrate the weaker response in cloud cover in Congo.

(Page: 7-8, Lines: 138-141) “The weaker warming response in the Congo may be driven by a smaller cloud response (that is, a smaller decrease in cloud cover, Supplementary Fig. 3) in Africa where the diurnal temperature range changes by a smaller amount as shown by previous simulations^{43,36}.”

Supplementary Figure 3. Biophysical impacts of idealized deforestation on total cloud cover fraction (%) in three tropical regions. Changes in mean annual cloud cover in a, South America, b, Africa and c, Southeast Asia. The changes were computed as the difference between the average of last 30 years from the LUMIP deforest-glob and piControl experiments (see Methods). Dotted area indicates the model agreement, with at least 6 models agreeing on the sign of the cloud responses. Information of the 8 models is listed in Table 1. Note that the cloud cover fraction is not available in the output for model BCC-CSM2-MR.

6. Line 293: How do physiological effect cause an increase in precipitation? I believe physiological effects cause a decrease in evapotranspiration. Do we know for sure it also causes an increase in precipitation and why? This issue should be discussed here.

Response: The reviewer is correct about the physiological effect of rising CO₂ causing a reduction local evapotranspiration. This is widely seen in CMIP5 and CMIP6 models. Yet, continental precipitation responses to these changes in surface biophysics are complex and vary within and across tropical continents, as shown by Kooperman et al. (2018) (this earlier paper is cited in our manuscript). Decreases in evapotranspiration could significantly reduce the precipitation in the Amazon basin, where precipitation recycling by means of evapotranspiration is important and surface evapotranspiration is essential for triggering convection in lowland forests near the coast (Langenbrunner et al. 2019). By contrast, precipitation over the Maritime Continent in tropical

Asia may rely more on non-local moisture transport. In tropical Asia, decreases in evapotranspiration over islands (and increases in sensible heating) may increase moisture convergence and precipitation over land (also see Fig. 2c,d in Kooperman et al. 2018).

To address the reviewer's concern regarding this issue, we have carefully modified the text to clarify the difference in these two mechanisms in driving the tropical precipitation response to increasing CO₂:

(Page: 16, Lines: 316-323) “In contrast, deforestation-driven decreases in rainfall in the western Congo and in some areas of tropical Asia may be partially offset by the radiative and physiological effects of rising CO₂, which may increase precipitation in these regions as a consequence of interactions between surface biophysical changes and regional atmospheric circulation⁴⁵. Nevertheless, the change of mean climate by deforestation (that is, less rainfall and warming), as revealed in our analysis of the LUMIP–CMIP6 simulations, still implies the increasing possibility of a lengthening dry season⁸², an increasing amplitude of extreme drought events⁸³, and a higher likelihood of wildfires^{5,10,15} in the tropics if deforestation continues in the future.”

References

Kooperman, G. J. et al. Forest response to rising CO₂ drives zonally asymmetric rainfall change over tropical land. *Nat. Clim. Chang.* **8**, 434-440 (2018).

Langenbrunner, B., Pritchard, M. S., Kooperman, G. J. & Randerson, J. T. Why does Amazon precipitation decrease when tropical forests respond to increasing CO₂?. *Earth. Future* **7**, 450-468 (2019).

7. Lines 302-303: I believe that the parameter γ does not care whether the CO₂ emissions are the result of fossil fuel burning or land use change. I agree that it does not include biophysical effect of land use change. This should be distinguished and clarified.

Response: We agree with the reviewer that this parameter γ refers to carbon cycle response to warming caused by increased CO₂ concentration, which could be contributed either by both fossil fuel burning or land-use change. We also agree that γ does not include the biophysical effect of land-use change, especially the deforestation in the tropics. Following the reviewer's suggestion, we modified the entire paragraph to more carefully distinguish and clarify the different concepts of γ .

(Page: 16-17, Lines: 328-349) “The carbon–climate feedback parameter γ was initially proposed to measure the carbon cycle response to climate warming^{56,70}, irrespective of whether the warming originates from fossil fuel emissions or carbon emissions from land-use change. In past applications, γ has been found to be negative (i.e., a loss of carbon to the atmosphere for a 1°C temperature increase) across tropical land ecosystems as a consequence of increases in ecosystem respiration and decreases in photosynthesis caused by warming⁸⁴. As far as we are aware, past work has not estimated γ driven by warming from the biophysical effects of tropical deforestation. This warming is fundamentally different because it is associated with large changes in precipitation and other land surface variables, including humidity and wind speed. Here we find that the deforestation-driven $\gamma_{AGB}^{def,biophys}$ (i.e., the term considering the biophysical warming effect of

deforestation on the carbon cycle in the tropics) could be twofold larger in the Amazon and Congo than that computed from the idealized CO₂ increasing experiment without consideration of land use and land cover change ($\gamma_{AGB}^{CO_2}$, as derived from the Coupled Climate–Carbon Cycle Model Intercomparison Project experiments (C4MIP) for CMIP6, ref. ⁷¹). This implies that the warming caused by biophysical effects of tropical deforestation has stronger impacts on nearby tropical terrestrial ecosystems than warming originating from global radiative forcing of the Earth system, once adjusted for the same change in temperature. Further analysis on the attribution of future tropical climate change to deforestation and CO₂ is needed for a better understanding of the role of tropical land use and land cover in the climate system. In future work, higher resolution model simulations may help to identify optimal locations for forest restoration efforts in order to offset precipitation declines from historic deforestation and maximize climate change mitigation from tropical reforestation efforts.”

(Page: 13, Lines: 264-265) “It should be noted that this $\gamma_{AGB}^{CO_2}$ calculation does not account for any biophysical climate effects of land use and land cover change⁷¹.”

8. Line 354: 20 million km². What fraction of the total forest cover is removed in this experiment? This fraction should be also mentioned to understand the magnitude of deforestation.

Response: Thanks. In the previous manuscript, we have already mentioned the fraction of the total forest cover being removed within our study areas in this experiment. (i.e., -44.7 ± 6.0% in Amazon, -38.7 ± 8.8% in Congo, and -31.2 ± 8.9% in tropical Asia) (Page: 20, Lines: 394-395, in the Methods).

9. Line 395: Why is the error smaller when the AGB is larger? This may be briefly discussed.

Response: According to the Product Validation & Algorithm Selection Report Version 2 of the ESA–CCI AGB data, we reported in the manuscript that the upper limit of relative error of AGB is 20% where AGB exceeds 50 Mg ha⁻¹ and a fixed error of 10 Mg ha⁻¹ where the AGB is below that limit. In other word, the upper limit of the AGB error would be larger (instead of being smaller) than 10 Mg ha⁻¹ when AGB is larger than 50 Mg ha⁻¹. We have carefully modified the statement to avoid any misunderstanding.

(Page: 22, Lines: 430-434) “The accuracy of ESA–CCI AGB has been improved when compared to the previous version (that is, AGB from GlobBiomass project, used in land surface model evaluation⁹⁰), with the upper limit of AGB relative error being 20% where AGB exceeds 50 Mg ha⁻¹ and a fixed error of 10 Mg ha⁻¹ where the AGB is below that limit (see Product Validation & Algorithm Selection Report Version 2 in <https://climate.esa.int/en/projects/biomass/key-documents/>).”

10. Line 398: Why “underestimate”?

Response: We have modified the texts in the Methods to clearly explain the reason for the possible underestimation of AGB in wet tropics.

(Page: 22, Lines: 435-436) “Nevertheless, ESA–CCI may still underestimate AGB in wet tropics because both L– and C– band backscatter data saturate at high AGB levels when AGB values keep increasing.”

11. Lines 417-420 and lines 431-435: This is contrary to what the paper Devaraju et al. 2015 PNAS finds. It is shown in this PNAS paper that remote high-latitude deforestation has larger effect on tropical precipitation than local deforestation.

Response: We thank the reviewer for the reminder of the remote effect from high-latitude deforestation on monsoon-region precipitation. While we appreciate the important finding of Devaraju et al. (2015) on highlighting the remote precipitation effects of high-latitude deforestation via shift in the Intertropical Convergence Zone, there are several key uncertainties in their study that may influence such conclusion and explain the difference between their findings and the results in our study.

First, Devaraju et al. (2015) applied an early version of the National Center for Atmospheric Research Community Atmosphere Model 5.0 (CAM5) coupled to the land surface model Community Land Model 4 (CLM4) (and a slab ocean model). CLM4 uses the Ball-Berry stomatal conductance model, which used a single slope parameter for all C3 plants. This model has been replaced by the Medlyn “empirical-optimal” conductance model in CLM5 (i.e., the version used in LUMIP experiments and our study) (Lawrence et al. 2019). With this optimized conductance model, we find that CLM5 could simulate a stronger evapotranspiration decline in response to local tropical deforestation. In other word, the earlier version of CLM (e.g., the one used in Devaraju et al. 2015) could underestimate the local deforestation effects on precipitation.

Second, according to the method in Devaraju et al. (2015), the simulated high-latitude deforestation fraction is close to 100%, which is larger enough to result in the southward shift in the intertropical convergence zone (ITCZ). By contrast, the magnitude of high-latitude deforestation fraction is much smaller in the LUMIP experiments (40%~60% for most models, see Fig. 1 in Boysen et al. 2020). It is not clear whether the position of the large-scale ITCZ would be significantly changed by a smaller deforestation fraction in the high-latitudes and in turn to affect the monsoon-region precipitation.

Third, the definition of regions of interest is substantially different between Devaraju et al. (2015) and our study. While we acknowledge that the high-latitude deforestation could have a significant remote effect on precipitation in the monsoon-regions, their remote effect is not as strong as the local effect of deforestation in tropical areas with high fraction of forests especially in central Amazon and Congo. Fig. 2 in Devaraju et al. (2015) clearly illustrates that the boreal and temperate large-scale deforestation does not have a significant remote effect on precipitation in central Amazon and Congo when compared to local deforestation in the tropics. In tropical Asia, the remote effects of high-latitude deforestation could be comparable to the local deforestation effects on precipitation.

To address the reviewer’s concern and clarify the important role of the remote effects by high-latitude deforestation on tropical precipitation as revealed by Devaraju et al. (2015), we have

carefully revised the texts and cite the important work by Devaraju et al. (2015) in the revised manuscript.

(Page: 23, Lines: 455-459) “Although tropical rainfall and air temperature may be perturbed significantly by deforestation in the extratropical regions (in particular, see remote effects by high-latitude deforestation in Devaraju et al.²⁶), most models agree that deforestation causes an averaged warming and decline in rainfall due to the large-scale decline in evapotranspiration within the tropics⁶⁵.”

(Page: 24, Lines: 472-475) “Nevertheless, more work is needed to systematically examine tropical and extratropical deforestation contributions (e.g., remote effects on tropical monsoon precipitation by high-latitude deforestation via shifting the intertropical convergence zone²⁶) to regional climate across the full suite of CMIP models.”

References

Devaraju, N., Bala, G., & Modak, A. Effects of large-scale deforestation on precipitation in the monsoon regions: Remote versus local effects. *Proc. Natl. Acad. Sci. U.S.A.* **112**, 3257-3262 (2015).

Lawrence, D. M. et al. The Community Land Model version 5: Description of new features, benchmarking, and impact of forcing uncertainty. *J. Adv. Model. Earth Syst.* **11**, 4245-4287 (2019).

Boysen, L. et al. Global climate response to idealized deforestation in CMIP6 models. *Biogeosciences* **17**, 5615-5638 (2020).

12. Line 494: It is shown that the AGB climate sensitivity varies with the rainfall background. Does it also vary with temperature background? This question should be briefly addressed here.

Response: We appreciate the reviewer’s rigorous evaluations. Using the same data as shown in Fig. 2 in the revised manuscript, we show below in Fig. R1a that the AGB climate sensitivity does not vary with the temperature background for the observation-based data (i.e., satellite-derived AGB and precipitation, and CRU temperature). The observational AGB sensitivity to MAP and MAT is consistent with the regression results using all grid cells in the tropics as shown in the Table 3 (i.e., AGB sensitivity to MAP: 3.4 Mg C ha⁻¹ per 100 mm yr⁻¹; AGB sensitivity to MAT: -0.32 Mg C ha⁻¹ per 1 °C). Although the AGB climate sensitivity seems to increase a little along with increasing temperature for the CMIP6 simulations (Fig. R1b), this would not influence our main conclusion which is drawn based on the AGB climate sensitivity using the observation-based data. We briefly clarified this in the revised manuscript.

(Page: 26, Lines: 530-533) “we used the moving window AGB climate sensitivity at different rainfall levels to estimate the carbon costs of tropical deforestation-driven climate change as the AGB climate sensitivity varies with the rainfall background (Fig. 2c).”

Fig. R1 Statistical sensitivity of aboveground biomass (AGB) to mean annual precipitation (MAP) and mean annual temperature (MAT) for **a**, observation-based data and **b**, CMIP6 simulations, respectively. The sensitivity of AGB to MAP and MAT was estimated from a multiple linear regression within each moving window spanning ± 5 °C and centered at each MAT level from 17 °C to 27 °C. Curve with dots indicating the regression with a significance of $P < 0.001$.

13. Lines 536-538: *I believe that the climate carbon feedback does not care whether the CO2 emissions are the result of fossil fuel burning or land use change. I agree that it does not include biophysical effect of land use change. This should be distinguished and clarified.*

Response: Please see our response to your similar comment above, we considerably modified the paragraph in the discussion and text elsewhere to account for this reviewer suggestion.

14. Table 2: *Absolute changes in precipitation and AGB are difficult to understand and appreciate and different regions may have different base values. Therefore, I suggest the authors to also list the % changes in these quantities in the table.*

Response: We agree that it's a good idea to also list relative changes in percentage in the table. This has been done by adding the relative precipitation change (%) in Table 2 in the revised manuscript. As for the AGB, the total forest AGB loss has been estimated by the product between the tree cover change (%) in LUMIP deforest-glob simulations after 50 years and the observational tree cover-aboveground biomass relationship as shown in Supplementary Fig. 7. Therefore, the relative changes in AGB should be identical to the relative change in tree cover in CMIP6 LUMIP experiments (already listed in Table 2). In the previous version of our manuscript, we have also provided the relative change in forest biomass (%) in Table 2.

15. Table 2: *the last 3 columns should clearly indicate that the quantities shown are AGB and not total forest carbon (which includes soil carbon)*

Response: Thanks. We have modified the last 3 columns to change “forest carbon” into “AGB” in Table 2 in the revised manuscript. To avoid misunderstanding of our results of the AGB rather than the total forest carbon, we have also changed the “vegetation carbon” into “AGB” in Fig. 3 and Fig. 4 in the revised manuscript.

16. Figure 1, top three panels: As indicated earlier, it would be useful to understand and appreciate if the % changes in precipitation are shown here or in a supplemental figure.

Response: We appreciate the constructive suggestion and have added the spatial pattern of change in precipitation in percentage (%) as a new supplementary figure S2.

Supplementary Figure 2. Same as Fig. 1a–c but in percentage (%) for the annual mean precipitation change in response to idealized deforestation in LUMIP experiments.

17. Figure 2, bottom 2 panels: Sensitivity is estimated for various base precipitation values. Why not also estimate the sensitivity for various background T values?

Response: We thank the reviewer for the rigorous evaluation. Similar to the response to your comment above, we have computed the AGB climate sensitivity for a varied temperature background as shown in Fig. R1. Please see our response to your similar comment above, we considerably modified the text in the Methods to account for this reviewer suggestion. We would respectfully prefer not to include this in the paper because for most tropical forests, the mean annual temperature range is fairly narrow (24°C–28°C) and the role of temperature in regulating the biomass is smaller, providing the potential for introduction of noise.

18. Figure 3, top 3 panels: It would be useful to understand and appreciate if the % changes in AGB are also shown here or in a supplemental figure.

Response: Thanks. The bottom 3 panels (i.e., Fig. 3d–f) are exactly the relative change in percentage (%) of the biophysical feedback of deforestation on AGB loss in the tropics. We have carefully modified the legend of Fig. 3 to reflect this.

19. Figure 4, top 3 panels: It would be useful to understand and appreciate if the % changes in AGB are also shown here or in a supplemental figure.

Response: We attempted to map the spatial pattern and found that the relative contribution from rainfall is generally over 80% (with color being quite homogeneous) of the total biophysical feedback (including both rainfall and temperature) to AGB loss both in Amazon and Congo basins. Mapping the relative contribution of rainfall makes no sense in tropical Asia as its total biophysical feedback to AGB is close to 0 (Fig. 3c). Therefore, instead of showing the relative contribution of

rainfall contribution, we added a sentence behind the legend of Fig. 4 to help the readers have a clear picture about the relative contributions the biophysical feedback to rainfall by deforestation.

(Page: 49, Lines: 904-906) “The relative contribution of precipitation change to biophysical AGB loss varies from 80% to 100% in the Amazon and from 85% to 100% in the Congo.”

20. *Figure S2: A panel that shows the differences between top and bottom panels should be also provided for understanding the differences between simulations and observations.*

Response: Thanks. We have provided another spatial pattern showing the differences in aboveground biomass between the CMIP6 model mean and the satellite-derived ESA CCI biomass in Supplementary Fig. 4 of the revised manuscript.

21. *Table S4, footnote: More parentheses should be used to clearly indicate which terms are in the denominator and which ones are in the numerator.*

Response: We have added more parentheses following the reviewer suggestion.

Reviewer #2

This study investigates the impacts of deforestation-induced climate change (precipitation and temperature) on carbon storage in remaining tropical forests using Earth system model simulations and satellite-derived aboveground biomass datasets. Previous relevant studies mainly analyze the losses of aboveground biomass directly from deforestation. This is a very interesting and important study to improve forest aboveground biomass loss under deforestation. I have some comments, which may help authors improve this manuscript.

Response: We appreciate the reviewer's recognition on the novelty and importance of our study in quantifying indirect carbon losses associated with deforestation as a consequence of warming and drying. We thank the reviewer for the constructive suggestions. Point-by-point responses are offered below.

Major comments:

I do not have much knowledge about these Earth system models. In this study, the model simulations have been used in estimating aboveground biomass losses from deforestation and deforestation-induced climate change, and precipitation and temperature changes caused by deforestation. What are the uncertainties of these aboveground biomass losses, and precipitation and temperature changes estimated by Earth System models, comparing to the long-term satellite observations or the field observations?

Response: There are two primary sources of uncertainty in our estimates of the feedback of deforestation-driven changes in climate on regional carbon stocks. The first is associated with Earth system model (ESM) estimates of the impact of a change in tropical forest tree cover on regional climate. The second is associated with the relationship between this climate change and its impact on aboveground forest biomass. The latter source of uncertainty has been addressed by using the observational biomass sensitivity to mean annual precipitation and temperature.

To address the former source of uncertainty associated with the ESMs, we first added the following text to the results section to comment on levels of model agreement more directly.

(Page: 7, Lines: 122-126) “Model agreement with respect to the direction of the precipitation response to deforestation is high in South America, with at least 6 of the 8 models showing decreases in precipitation across most of the Amazon. Model agreement is lower in the eastern part of the Congo and across tropical Asia, where the magnitude of the multi-model mean change is also smaller relative to background precipitation levels.”

For temperature, our existing text draws the reader's attention to the model-to-model differences:

(Page: 7, Lines: 128-131) “In response to idealized deforestation, mean annual air temperature increases significantly in the Amazon by 0.5 ± 0.5 °C for the multi-model mean, with smaller and more variable cross-model responses in the Congo (0.1 ± 0.5 °C) and tropical Asia (-0.1 ± 0.2 °C) (Fig. 1d-f and Table 2).”

More broadly, we acknowledge that there are uncertainties associated with the ESM simulations, which originate from different representations of carbon and surface energy fluxes in the individual

land surface models in each ESM, as well as different model representations of the atmosphere and its interactions with land surface processes and ocean dynamics.

First, the aboveground biomass losses due to deforestation are accounted for by the land surface vegetation models used in ESMs, but the simulated biomass losses may be different across different models even under the same experiment protocol such as the LUMIP idealized deforestation simulations (deforest–glob, Supplementary Fig. 1, left column), depending on the different assumptions of initial forest cover fraction and treatments of land-use change in specific models (Boysen et al. 2020). Instead of directly using the models simulated aboveground biomass, the estimation in our study is, in fact, based on the models simulated tree cover losses (i.e., more consistent across models than the biomass losses, Supplementary Fig. 1, right column) and the satellite-observed relationship between tree cover fraction (from MODIS) and aboveground biomass (from ESA–CCI). In other words, the aboveground biomass losses due to deforestation estimated in our study are constrained by the satellite observed biomass–tree cover relationship, which, to the best of our knowledge, reduces the ESMs’ uncertainties in estimating the deforestation-caused aboveground biomass losses.

Second, we admit that there are few direct observations (either from satellite or field investigations) that can be compared with the precipitation and temperature changes due to deforestation as estimated from the LUMIP deforest–glob experiments used in our study. However, despite this fact that there are no direct observations to be compared to our results, we recognize that there is increasing evidence from the satellite observations that support the biophysical mechanisms (i.e., increased albedo and declined evapotranspiration) and biophysical climate effects (i.e., warming and declined precipitation) of deforestation, consistent with the existing mechanisms in the ESMs. For instance, Li et al. (2015) applied a space for time method to show tropical forests have a strong cooling effect throughout the year via a stronger evapotranspiration and a higher albedo. Leite-Filho et al. (2021) confirmed the ESM simulated rainfall decline with deforestation in the southern Brazilian Amazon using the satellite observations. Inspired by the reviewer’s comment, we realize that reducing the uncertainties of model simulated temperature and rainfall changes due to deforestation requires further development of the process-based evaluation framework of the biophysical climate effects of deforestation for reconciling the ESMs simulations with the satellite/field observations (e.g., similar to the one developed by Duveiller et al. 2018 for evaluating vegetation cover influence on surface energy balance).

To address the reviewer’s concerns about the ESMs’ uncertainties and highlight the importance to develop a process-based evaluation framework of the biophysical climate effects of deforestation, we have carefully modified the text to reflect the above key points in the revised manuscript:

(Page: 11, Lines: 213-219) “The satellite observations indicated that AGB decreases by about 19–22 Mg C ha⁻¹ per 10% decrease in tree cover fraction in the tropics (Supplementary Fig. 7). Using this relationship derived from the observations and the tree cover fraction changes from the LUMIP models, we estimate that direct AGB carbon losses due to the deforestation in the idealized deforestation experiments were 98.3 ± 13.2 Mg C ha⁻¹, 75.5 ± 16.7 Mg C ha⁻¹, and 62.4 ± 17.8 Mg C ha⁻¹ in the Amazon, Congo, and tropical Asia, respectively (Table 2).”

(Page: 18, Lines: 362-369, in the discussion) “This may include further effort to develop a process-based evaluation framework for quantifying the impact of deforestation on regional climate that reconciles predictions from ESMs with long-term trends from satellites and field observations, building on the framework developed by Duveiller et al.⁸⁶ for evaluating the influence of vegetation cover on surface energy exchange. Additional improvements of ESMs to better represent forest mortality, convection, and precipitation in tropical climate simulations also would help us more accurately assess the carbon costs of deforestation and evaluate their role in the changing climate system.”

References

Boysen, L. et al. Global climate response to idealized deforestation in CMIP6 models. *Biogeosciences* **17**, 5615-5638 (2020).

Li, Y., Zhao, M., Motesharrei, S., Mu, Q., Kalnay, E., & Li, S. Local cooling and warming effects of forests based on satellite observations. *Nat Commun.* **6**, 1-8 (2015).

Leite-Filho, A. T., Soares-Filho, B. S., Davis, J. L., Abrahão, G. M., & Börner, J. Deforestation reduces rainfall and agricultural revenues in the Brazilian Amazon. *Nat. Commun.* **12**, 1-7 (2021).

Duveiller, G. et al. Biophysics and vegetation cover change: a process-based evaluation framework for confronting land surface models with satellite observations. *Earth Syst. Sci. Data* **10**, 1265-1279 (2018).

Minor comments:

Line 44-46: Lee et al paper is not about the tropical forest and may not be cited to support this sentence.

Response: Thanks for pointing out this. We have replaced this reference with Li et al. (2015), who studied the global biophysical temperature effects of forests based on satellite observations. We think this one is more appropriate as it covers the tropical forest.

Reference

Li, Y., Zhao, M., Motesharrei, S., Mu, Q., Kalnay, E., & Li, S. Local cooling and warming effects of forests based on satellite observations. *Nat Commun.* **6**, 1-8 (2015).

Line 98: The definition of carbon cost can be moved from the Results section (Line 204) to here.

Response: We have moved the definition of biophysical carbon cost from the Results section to the last paragraph of the Introduction following the reviewer suggestion (Page: 6, Lines: 99-100).

Line 101: What is the meaning of gamma? It is not clear to readers.

Response: We have added a more complete explanation of gamma (in parentheses in text below) in the revised manuscript.

(Page: 6, Lines: 101-104) “We also report the carbon–climate feedback parameter, gamma (defined as the cumulative carbon loss at each location for a 1°C increase in surface air temperature)⁵⁶ driven solely by the biophysical climate effect of tropical deforestation and compare it to more traditional estimates of gamma derived from radiative effects of increasing CO₂.”

Line 142: When analyzing the sensitivity of tropical aboveground biomass to climate, why did you use a moving window (± 500 mm) for precipitation?

Response: We use a moving window for precipitation to accurately calculate and better represent the aboveground biomass climate sensitivity in tropical forests on different tropical continents. As shown by Fig. 2c, the rainfall sensitivity and temperature sensitivity vary for different precipitation background levels, with the biomass being more sensitive to climate in Amazon and Congo (at lower initial background levels) than that in tropical Asia (higher initial background levels). By combining this segmented biomass climate sensitivity at different precipitation background with the ESMs simulated rainfall and temperature changes due to deforestation, we could estimate a more accurate biomass losses owing to the deforestation-induced climate change at the three tropical continents.

We added the following text to make it clearer why we used a moving window:

(Page: 10, Lines: 182-183) “To tailor a climate-biomass statistical model for use in wetter areas where tropical forests are dominant, we used a moving window on climatological annual rainfall (see Methods).”

(Page: 26, Lines: 530-536, in the Methods) “Instead of using a unified AGB sensitivity to MAP and MAT (shown in Table 3), we used the moving window AGB climate sensitivity at different rainfall levels to estimate the carbon costs of tropical deforestation–driven climate change as the AGB climate sensitivity varies with the rainfall background (Fig. 2c). When applying this moving window approach, we inferred the AGB climate sensitivity at each grid cell by its background rainfall level, which was computed as the GPCC⁹¹ climatological precipitation average plus the deforestation-induced change in relative rainfall (%) from the LUMIP experiments.”

Line 172: The land cover map may be needed in supplementary materials to help readers understand the tropics' spatial distribution of forest and savanna.

Response: We added a land cover map derived from the MODIS satellite during the year 2017, which corresponds to the year of the ESA–CCI biomass map as a supplementary figure to address this reviewer suggestion and cite it in the main text.

(Page: 9, Lines: 176-180) “When comparing our findings to those in the previous study, it is important to note our regression is also derived from non–forest biomes in the tropics, including savannas and grasslands (Supplementary Fig. 5). Therefore, the climate gradient that we consider

is larger, and rainfall therefore has a more significant role in defining the transition from forests to savannas.”

Supplementary Figure 5. Tropical land cover map showing land cover type for each 1-degree grid cell within 23°S–23°N. In our analysis, desert regions with mean annual precipitation less than 100 mm yr⁻¹ and grid cells with a land fraction less than 50% were masked when computing the aboveground biomass (AGB) climate sensitivity to avoid use of observations where uncertainties are high. The land cover map was derived from MODIS Land Cover Type (MCD12C1) during the year 2017 (consistent with the ESA–CCI AGB map), with the International Geosphere-Biosphere Programme (IGBP) class system as shown above. The MCD12C1 map has a spatial resolution of 0.05 degree and has been resampled into 1-degree map with each 1-degree grid cell in the map showing the most abundant land cover type.

REVIEWER COMMENTS

Reviewer #2 (Remarks to the Author):

The authors have addressed most of my concerns. I would like to suggest accepting this paper for publication.

We wish to thank the two reviewers for their positive feedback and constructive suggestions. And we appreciate their time and efforts on this manuscript. The revised manuscript only has tiny edits according to the editorial requests regarding altering the rainbow color schemes in Fig.1 and Fig. 5, as well as linking our Github repository to Zenodo to obtain a citable DOI in the code availability statement.

Reviewers' comments:

Reviewer #2 (Remarks to the Author):

The authors have addressed most of my concerns. I would like to suggest accepting this paper for publication.

Response: We thank the Referee for the work and time on reviewing this manuscript. We appreciate that the Referee is satisfied with our revisions.